# Towards the Prediction of Tensile Properties in Automotive Cast Parts Manufactured by LPDC with the A356.2 Alloy

Jon Ander Santamaría [1,*], Jon Sertucha [2] , Alberto Redondo [1], Ibon Lizarralde [2], Edurne Ochoa de Zabalegui [1] and Patxi Rodríguez [1]

1    Edertek S. Coop., Garaia Berrikuntza Gunea, Isasi Kalea 6, PK 19, E-20500 Arrasate-Mondragón, Gipuzkoa, Spain; a.redondo@edertek.es (A.R.); e.otxoa@edertek.es (E.O.d.Z.); p.rodriguez@edertek.es (P.R.)
2    Fundación Azterlan, Basque Research Technological Alliance, Aliendalde Auzunea 6, E-48200 Durango, Bizkaia, Spain; jsertucha@azterlan.es (J.S.); ilizarralde@azterlan.es (I.L.)
*    Correspondence: j.santamaria@edertek.es ; Tel.: +34-94-3038867

**Abstract:** Aluminum-silicon-magnesium alloys are commonly used in the automotive industry to produce structural components. Among usual quality controls of produced castings, microstructure characterization and determination of mechanical properties are the most critical aspects. However, important problems can be found when measuring mechanical properties in those areas of castings with geometrical limitations. In this investigation, a set of A356 alloys have been prepared and then used to manufacture test castings and automotive castings in a laboratory and in industrial conditions, respectively, using Low Pressure Die Casting (LPDC) technology. Test castings were used to predict secondary dendritic arm spacing (SDAS) by using thermal parameters obtained from experimental cooling curves. The results have been then compared to the ones found in the literature and improved methods for estimating SDAS from cooling curves have been developed. In a subsequent step, these methodologies have been checked with different industrial castings by using simulated cooling curves and experimentally measured SDAS values. Finally, the calculated SDAS values together with the Mg contents present in A356 alloys and the temperature and aging time data have been used to develop new models so as to predict the tensile properties in different areas of a given casting prototype. These developed models allow casters and designers predicting tensile properties in selected areas of a given prototype casting even during design and simulation steps and considering the processing variables expected in a given foundry plant. The structures of these new models have been described and experimentally validated using different processing conditions.

**Keywords:** Al-Si-Mg alloys; mechanical properties; solidification parameters; SDAS; automotive castings; LPDC; prediction models

## 1. Introduction

AlSi7Mg0.3 alloys, also known as A356.2, are widely used to produce safety cast components in automotive and aerospace industries due to their excellent castability, corrosion resistance, light weight, and to the important development of these alloys in terms of improved mechanical properties by controlling their structural characteristics and performed heat treatment—T6 being the most usual one [1–3]. For instance, the A356.2 alloy is the most common one for producing wheels, knuckles, wheel-carriers, and subframe components in low-pressure die casting (LPDC) technology [4,5]. LPDC is a common process producing high-quality castings resulting from two main characteristics: well-controlled low turbulence filling from the bottom of the mold cavity when compared to the turbulent flow associated to high-pressure die casting (HPDC), thus preventing melt oxidation and the presence of biofilms, oxides, and gas entrapment in the cast part related to turbulent flows and, on the other hand, a fast and directional solidification from the top to the bottom occurs once the cavity is filled, maintaining enough pressure during the

solidification step to avoid the formation of shrinkage porosities, resulting in a fine-grained and homogeneous microstructure.

Mechanical properties of Al-Si cast alloys are strongly dependent on the local microstructural features such as grain size, secondary dendrite arm spacing (SDAS), the presence of gas or shrinkage porosities, eutectic-silicon morphology, and Fe-bearing intermetallic phases size and morphology [6–9]. Adding master alloys containing Sr or $TiB_2$ particles is a well-established methodology in cast aluminum foundries to modify the Si-eutectic morphology from coarse plate-like forms into fine and rounded ones, and to refine the primary aluminum phase thus producing fine and uniform grains distributions, respectively [10–12]. Additionally, solidification rate is one of the most important processing variables which makes more evident its effects of the Si-eutectic modification, strongly affecting both the microstructure and mechanical properties of aluminum castings [11,13,14]. The reported benefits of increasing cooling rate are the reduction of the grain size and SDAS, the formation of eutectic-silicon with a rounded morphology which leads to more homogeneous structures without large eutectic silicon islands (eutectic macrosegregation), a low porosity formation and the inhibition of coarsening and growth of Fe-rich intermetallic phases.

It is well-known that tensile properties and more precisely ductility of Al-Si cast alloys with the absence of metallurgical defects are strongly affected by the SDAS. This parameter is a direct consequence of the thermal gradients originated between liquidus and eutectic temperature values during aluminum solidification. However, the published data on mechanical properties are generally obtained from test castings manufactured in laboratories and/or separately produced with the castings [6]. These test castings conditions are different from those obtained from industrial castings produced by LPDC, as in this work; thus, relevant deviations can be found when estimating the mechanical properties using the models available in the literature. On the other hand, tensile strength of Al-Si-Mg cast alloys mainly depends on their Mg content and on the T6 heat-treatment parameters, since the essential hardening mechanism is the strengthening by $Mg_2Si$ precipitation [15–17]. Thus, the strength and ductility of these alloys can be controlled by varying the applied heat treatment.

In the present work, the experimental models reported by Chen et al. [18] for determining SDAS values from cooling curves have been compared to those results obtained from cylindrical test castings, and a series of improved equations have been then applied to different cases involving automotive castings produced using a Sr-modified and grain refined A356.2 alloy by LPDC. Then, the parts were subjected to different T6 heat treatments. Although analogous results are obtained for the models based on cooling rate, the one reported by Chen et al. based on SDAS determination through solidification time showed important deviations on the industrial scale. Thus, improved calculations have been developed to determine proper SDAS values using test castings and industrial castings produced by LPDC. Once obtained the SDAS values, Mg content of the A356.2 alloy and the T6 aging parameters (temperature and time) have been additionally considered to create new models for predicting tensile properties, i.e., yield strength, ultimate tensile strength, and elongation in selected areas of castings. The innovative models developed in the present investigation have been described and then validated using industrial castings.

## 2. Materials and Methods

Two different groups of castings were considered in the present study: on one hand cylindrical test castings were produced in a laboratory facility. On the other hand, a test arm casting together with eight industrial castings with different geometries were manufactured in two LPDC aluminum foundry plants. For the first group, eight sets with five different sized cylindrical test castings each, and with same diameter and height (see Figure 1) were manufactured according to the experimental methodology detailed in reference [19]. K-type thermocouples (Heraeus Electro-Nite España, S.L., Asturias. Spain) were placed in the center of all these test castings to record the corresponding cooling

curves and metallographic inspections were then made in these central areas. The analyses of the cooling curves and the metallographic inspections allowed determining the solidification time ($t_{SL}$), the cooling rate (CR), the SDAS values calculated according to the equations reported by Chen et al. [18], and the experimental SDAS values obtained from the microstructural studies.

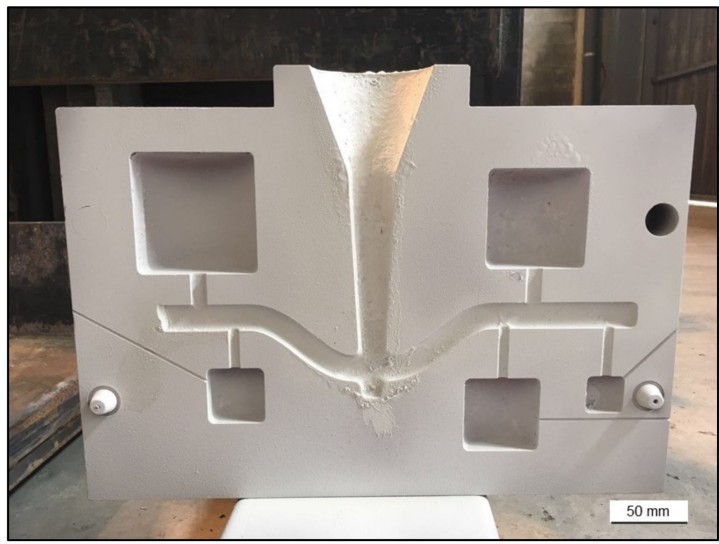

**Figure 1.** Lay out and the different cavities used to produce the cylindrical test castings using a die mold.

The second group was composed of eight industrial castings which were all automotive steering knuckles (SK) with different geometries and one test arm (TA) as listed in Table 1. All these castings were manufactured by LPDC with A356.2 alloys with the compositions shown in Table 2. The models for predicting the mechanical properties were developed with the results obtained from castings SK-1 to SK-6 and those from castings TA which underwent 13 different T6 heat-treatment conditions. These models were finally validated with the data obtained from castings SK-7 and SK-8 which underwent two different T6 heat-treatments with different Mg contents. To carry out all these heat-treatments, an important number of castings TA, SK-7 and SK-8 were needed, and different melt batches were thus used in each case to manufacture them as indicated in Table 2.

**Table 1.** Different industrial castings and number of sections used in each case.

| Casting Ref. | Weight (kg) | Analyzed Areas | Models Step |
|---|---|---|---|
| SK-1 | 4.17 | 4 | |
| SK-2 | 2.93 | 3 | |
| SK-3 | 2.98 | 3 | |
| SK-4 | 2.48 | 3 | Development |
| SK-5 | 2.80 | 3 | |
| SK-6 | 3.96 | 2 | |
| TA | 2.40 | 1 | |
| SK-7 | 3.24 | 5 | Validation |
| SK-8 | 4.28 | 2 | |

**Table 2.** Chemical composition of the melt batches prepared to manufacture the industrial castings (wt.%, Al balance) and aging parameters used in each case (to be complemented with Table 3).

| Casting | Si | Fe | Mg | Ti | Mn | Sr | $T_{aging}$ (°C) | $t_{aging}$ (min) |
|---------|-----|------|------|------|-------|-------|---------|----------|
| SK-1 | 7.35 | 0.09 | 0.35 | 0.10 | 0.002 | 0.018 | 165 | 280 |
| SK-2 | 7.26 | 0.11 | 0.33 | 0.13 | 0.008 | 0.020 | | |
| SK-3 | 7.20 | 0.12 | 0.34 | 0.12 | 0.002 | 0.018 | | |
| SK-4 | 7.18 | 0.09 | 0.35 | 0.12 | 0.006 | 0.022 | 155 | 360 |
| SK-5 | 7.45 | 0.11 | 0.38 | 0.10 | 0.005 | 0.021 | | |
| SK-6 | 7.30 | 0.09 | 0.36 | 0.12 | 0.005 | 0.019 | | |
| TA batch 1 | 7.23 | 0.12 | 0.35 | 0.11 | 0.004 | 0.016 | see Table 3 | |
| TA batch 2 | 7.20 | 0.11 | 0.38 | 0.10 | 0.003 | 0.013 | | |
| SK-7 batch 1 | 7.26 | 0.09 | 0.34 | 0.10 | 0.002 | 0.012 | 165 | 280/360 |
| SK-7 batch 2 | 7.32 | 0.10 | 0.42 | 0.12 | 0.003 | 0.011 | | |
| SK-8 batch 1 | 7.22 | 0.11 | 0.34 | 0.11 | 0.008 | 0.021 | | |
| SK-8 batch 2 | 7.31 | 0.10 | 0.33 | 0.11 | 0.009 | 0.018 | 155 | 360 |
| SK-8 batch 3 | 7.10 | 0.10 | 0.31 | 0.10 | 0.011 | 0.016 | | |
| SK-8 batch 4 | 7.23 | 0.11 | 0.30 | 0.10 | 0.012 | 0.018 | | |

**Table 3.** Parameters used during the aging step (TA castings).

| $T_{aging}$ (°C) | $t_{aging}$ (min) | | | | | | |
|---------|-----|-----|-----|-----|-----|-----|-----|
| 165 | 50 | 100 | 200 | 280 | 400 | 460 | — |
| 155 | 50 | 100 | 200 | 360 | 480 | 510 | 600 |

Melts were produced by introducing aluminum ingots (wt.%, 7.02 Si, 0.11 Fe, 0.31 Mg, 0.09 Ti, Al balance) in a gas fired tower furnace (Insertec, Bizkaia, Spain) with a melting capacity of about 2 t/h. After completing the melting step, the liquid alloys were transferred from the furnace to a 2500 kg ladle where the Mg contents were adjusted into the range 0.35–0.42 wt.% by adding 100 g pure Mg ingots. In the following step, melts were refined with an addition of 0.2% of Al5Ti1B master alloy rods (wt.%, 4.5 Ti, 0.9 B, 0.12 Fe, 0.16 Si, 0.04 V, Al balance) for grain refinement. For eutectic modification, the Sr content of the melts was increased up to 0.025 wt.% by adding AlSr10 master alloy rods (wt.%, 10.6 Sr, 0.13 Fe, 0.03 Si, 0.02 Ca, Al balance). Then, the alloys were degassed with 25 l/min of nitrogen gas (>99.999 wt.%) injected by using a graphite rotor for 10 min (Figure 2). Then, melts were cleaned to remove the slag formed and a sample was obtained to check the composition. At this point, tests for determining the density index (estimation of the amount of gas dissolved in the melts) were made keeping this parameter below 5% to validate the batches in all cases. Finally, samples were obtained from each melt batch to determine the final composition with the Spark Optical Emission spectrometers Spectrolab LAVM11 (Spectro Hispania S.L., Bizkaia, Spain).

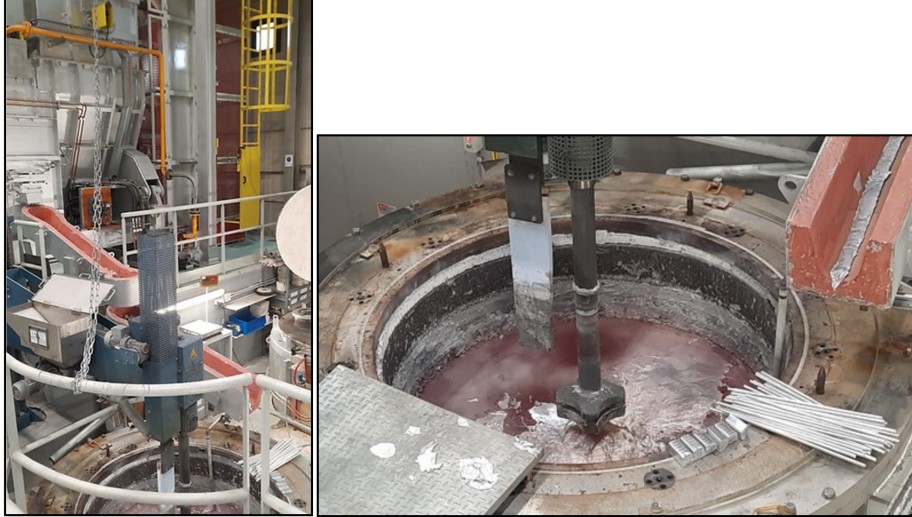

**Figure 2.** Gas tower furnace used for melting process (image on the **left**) and ladle used for degassing, adjustment of composition and addition of AlSr10 and Al5Ti1B master alloys (image on the **right**).

Once validated, the melt batches were transferred in the same ladle to a LPDC facility with a die cooling management system. The industrial castings were manufactured at 705 ± 10 °C into metallic permanent molds and cooled down with a closed water-cooling circuit following the usual thermal management strategies used in normal production (the temperature of the die molds achieved 515 ± 15 °C during the filling step and it was cooled down at about 320 ± 15 °C just before extracting the castings from the die).

After removing the castings from the metallic molds, they were T6 heat-treated using two gas-fired semicontinuous furnaces. The solubilization steps were performed at 540 °C for 6 h and the castings were then quenched in water at 40 °C. This quenching process required delay periods lower than 28 s in all cases. The already heat-treated and quenched castings were directly aged preventing natural aging in a second gas-fired semicontinuous furnace.

Aging temperature ($T_{aging}$) and time ($t_{aging}$) values were 165 °C and 280 min for castings SK-1 while these parameters were 155 °C and 360 min for castings SK-2 to SK-6. The comparatively high aging temperature and low aging time used for castings SK-1 was due to the higher tensile requirements of these cast parts. On the other hand, castings SK-7 used in the validation step were produced with two different Mg contents (0.34 wt.% and 0.42 wt.%, see Table 2) and both compositions were then aged at 165 °C for 280 min and for 360 min. Four batches were prepared to produce SK-8 castings with increasing Mg contents (from 0.30 wt.% to 0.34 wt.%, see Table 2) and aged at 155 °C for 360 min. For castings TA, they were used to study the effect of aging conditions on mechanical properties so different temperatures and aging times ($t_{aging}$) were used as shown in Table 3. The temperature accuracy of the furnaces was ±5 °C and ±3 °C for the solubilization and aging steps respectively.

All tensile specimens had a circular cross-section with cylindrical unthreaded ends, and they were machined out from the selected areas of the industrial castings (one specimen obtained per area and casting, see Figure 6). Depending on the dimension of the casting section, the specimens had a calibrated diameter of 4, 5, or 6 mm with 20, 25, or 30 mm in length, respectively (type A specimens in DIN 50,125). Thus, the ultimate tensile strength (UTS), the elongation (A) and the yield strength (YS) values were measured using a Hoytom HM-D 50 KN tensile testing equipment (Hoytom, Bizkaia, Spain) at a speed of 1 mm/min in the elastic range and of 5 mm/min in the plastic zone. A minimum number of 5 tensile specimens were tested per analyzed area of each casting reference included in Table 1.

One part of all broken tensile specimens was then used for metallographic inspections also including the fracture surface. After polishing the corresponding surfaces of all these

samples, they were used without any etching to determine the experimental SDAS values by the mean linear intercepted method [20]. 3 fields with a magnification of 100x were used per sample to obtain the average value of SDAS.

The filling and cooling processes for each industrial casting were simulated with the Magma 5.4.2 software (Magma Giessereitechnologie GmbH, Aachen, Germany). Nodes were positioned in the selected areas so as to obtain the corresponding cooling curves. The temperature versus time data thus acquired were then used to determine the characteristic temperature parameters such as solidification time and the cooling rate.

## 3. Results and Discussion

### 3.1. Cylindrical Test Castings

Figure 3 shows the different cooling curves recorded from one of the die molds poured to produce the five test castings with diameter and height of 24, 36, 48, 60, and 72 mm [19]. As expected, it is observed that by increasing the diameter of the test casting (lowering cooling rate values) the length of the solidification range increases. These curves together with the corresponding first derivative ones have been used to identify the temperature and time values for the initial mass nucleation of $\alpha$-Al ($T_{Al}$ and $t_{Al}$, respectively) and for the Al-Si eutectic ($T_{Eutectic}$ and $t_{Eutectic}$, respectively) following the criteria adopted by Chen et al. [18]. As an example, these characteristic solidification parameters have been indicated in the curves shown in Figure 4 for the test casting with 48 mm in diameter. In each cooling curve, the criterion t = 0 has been adopted at the maximum temperature registered by the thermocouple.

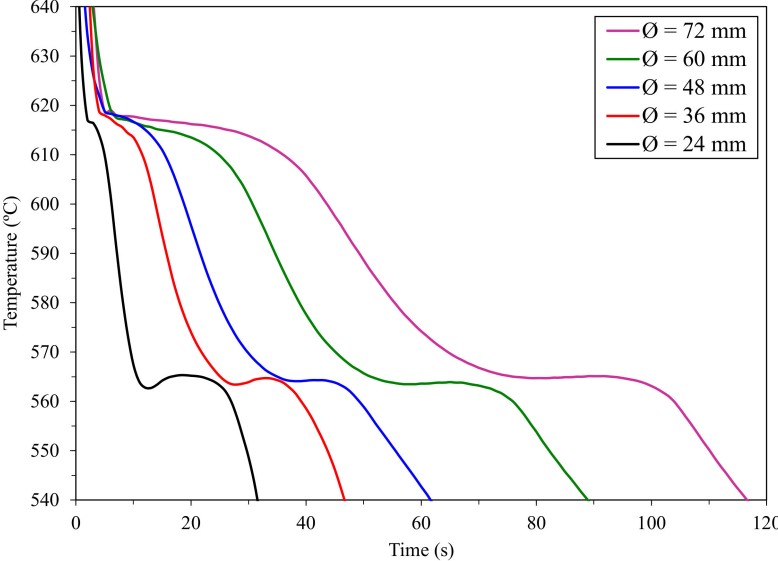

**Figure 3.** Different cooling curves obtained from the cylindrical test castings produced in one of the sets (data from [19]).

SDAS values measured at any location in a casting can be related to the local solidification time and the cooling rate at that point [19–21]. In the present work, $t_{SL}$ parameter has been determined from the cooling curves as $t_{SL} = t_{Eutectic} - t_{AL}$ while CR values have been obtained according to Equation (1) which represents the average cooling rate of primary $\alpha$-Al dendritic growth [18].

$$CR \, (°C/s) = \frac{T_{Al} - T_{Eutectic}}{t_{Eutectic} - t_{Al}} \qquad (1)$$

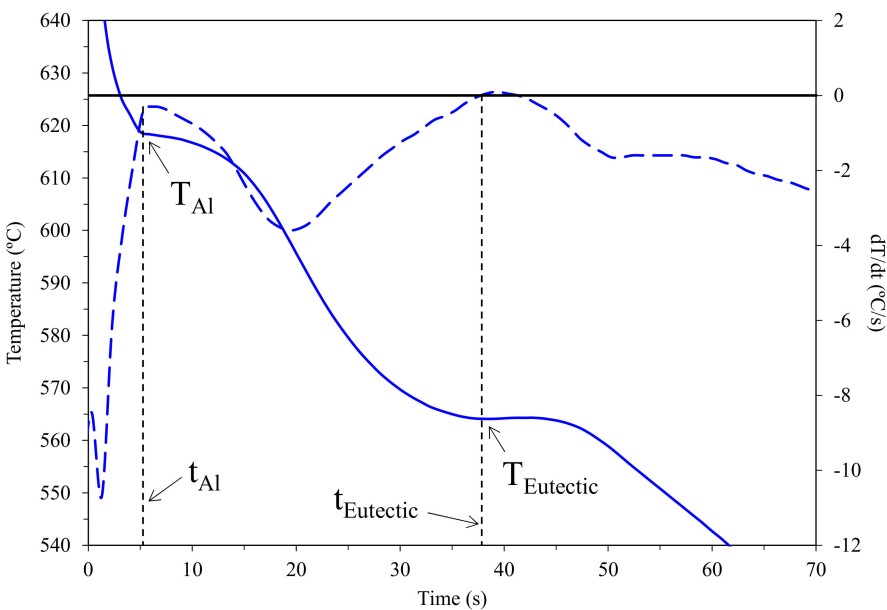

**Figure 4.** Characteristic solidification parameters in the cooling curve obtained from the test casting with 48 mm in diameter. The solid line corresponds to the cooling curve and the interrupted line is the first derivative curve.

A second cooling rate parameter ($CR_{635}$) has been also determined from cooling curves [19] which corresponds to the ratio between the temperature range defined by five values before and after 635 °C value, under the time range defined by such a temperature interval. The average values of the characteristic parameters determined on the cooling curves recorded from each test casting size are listed in Table 4. On the other hand, Table 5 includes the average values of $t_{SL}$, CR, $CR_{635}$ and SDAS, this last corresponding to those values metallographically determined [19].

**Table 4.** Average values of the solidification parameters determined from the cooling curves (standard deviation values are shown between brackets).

| Diameter (mm) | $T_{Al}$ (°C) | $t_{Al}$ (s) | $T_{Eutectic}$ (°C) | $t_{Eutectic}$ (s) |
|---|---|---|---|---|
| 24 | 614.6 (3.3) | 2.8 (0.4) | 562.8 (4.2) | 13.4 (1.7) |
| 36 | 615.8 (2.4) | 6.0 (1.1) | 564.9 (3.7) | 28.9 (2.4) |
| 48 | 617.3 (1.3) | 7.7 (1.6) | 565.5 (4.0) | 39.5 (5.2) |
| 60 | 617.7 (2.0) | 9.4 (1.9) | 566.6 (4.7) | 58.1 (4.3) |
| 72 | 618.2 (1.7) | 9.9 (1.4) | 567.3 (3.4) | 78.1 (4.6) |

**Table 5.** Average values of $t_{SL}$, CR, $CR_{635}$ and the experimentally measured SDAS for each test casting section (standard deviation values are shown between brackets).

| Diameter (mm) | $t_{SL}$ (s) | CR (°C/s) | $CR_{635}$ (°C/s) | SDAS (μm) |
|---|---|---|---|---|
| 24 | 10.7 (1.8) | 5.0 (0.8) | 31.5 (2.8) | 22.9 (0.3) |
| 36 | 22.9 (2.3) | 2.2 (0.2) | 19.3 (1.3) | 28.3 (0.4) |
| 48 | 31.9 (5.2) | 1.7 (0.2) | 15.8 (1.4) | 30.4 (0.7) |
| 60 | 48.7 (3.9) | 1.0 (0.1) | 8.8 (0.3) | 36.7 (1.3) |
| 72 | 68.2 (4.7) | 0.7 (0.1) | 5.1 (2.1) | 42.2 (0.5) |

The experimental SDAS data listed in Table 5 have been plotted versus $t_{SL}$ and CR in Figure 5 (open symbols) together with the results reported by Chen et al. [18] (solid

symbols). In both cases, a good fit of both groups of data is found though the agreement seems better for CR than for $t_{SL}$. It is also observed that the range of the solidification time covered in this work is shorter than that obtained in the study by Chen et al. These authors cast step castings with geometric moduli in the range 0.14–0.74 cm using a sand mold. The moduli of the cylindrical test castings present a higher range from 0.80 cm to 2.40 cm than those determined for the different sections in the step casting. This fact is due to cylindrical castings were cast in a die mold (see Figure 1) equipped with a cooling system which always originated $t_{SL}$ values lower than 70 s.

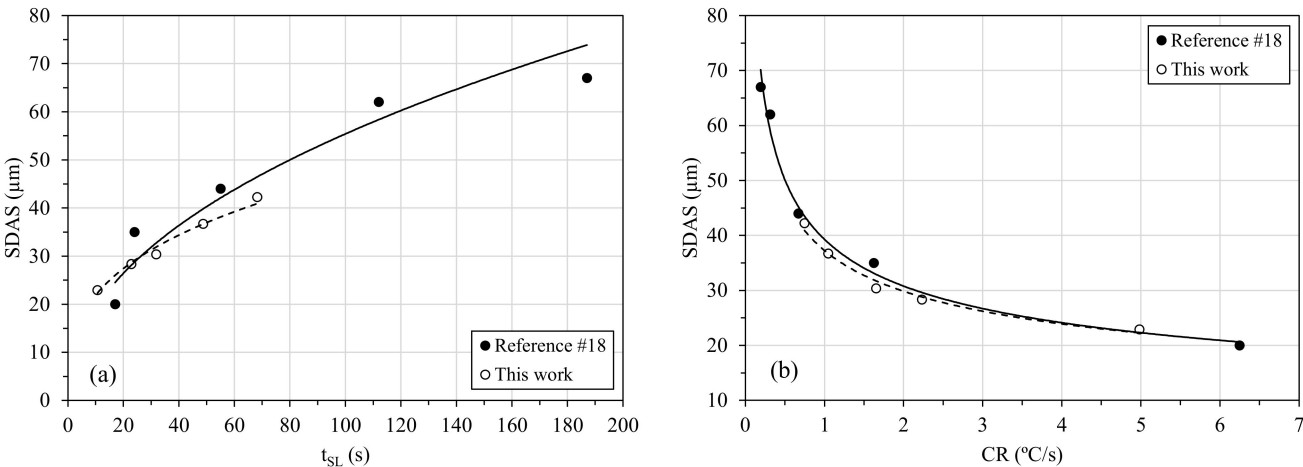

**Figure 5.** Evolution of experimental SDAS of the cylindrical test castings with (**a**) $t_{SL}$ and (**b**) CR. Black dots are obtained using data from [18].

According to the results shown in Figure 5 and to the data reported in the literature SDAS depends on solidification time and on cooling rate parameters following exponential correlations like $SDAS = K_1 \cdot (t_{SL})^{n_1}$ or $SDAS = K_2 \cdot (CR)^{n_2}$, where K and n coefficients mainly depend on alloy composition and on casting geometry and cooling conditions [18,21]. Considering the good agreements shown in Figure 5 between SDAS and $t_{SL}$ or CR, the data included in Table 5 lead to Equations (2) and (3) which represent the best least square fit for each of these two parameters obtained from cooling curves.

$$SDAS \ (\mu m) = 10.314 \cdot (t_{SL})^{0.326} \ R^2 = 0.981 \qquad (2)$$

$$SDAS \ (\mu m) = 37.271 \cdot (CR)^{-0.320} \ R^2 = 0.982 \qquad (3)$$

As expected, the coefficients K and n in Equations (2) and (3) are similar to those obtained by Chen et al. [18] from their own data. It is worthy to note here that a good correlation has been also found between SDAS and $CR_{635}$, as it was already documented in reference [19]. That calculation has been made again in this work leading to Equation (4):

$$SDAS \ (\mu m) = 74.737 \cdot (CR_{635})^{-0.334} \ R^2 = 0.990 \qquad (4)$$

*3.2. Industrial Castings*

As listed in Table 1, eight SK castings have been used in this section to determine the experimental SDAS values in different areas as indicated in Figure 6. All these castings together with casting TA have been also used to determine tensile properties in these areas. As mentioned in Section 2, castings SK-1 to SK-6 with similar T6 treatments and casting TA with different aging processes (see Table 3) were used into a preliminary group to develop models for predicting tensile properties while castings SK-7 and SK-8 were finally used during the validation step.

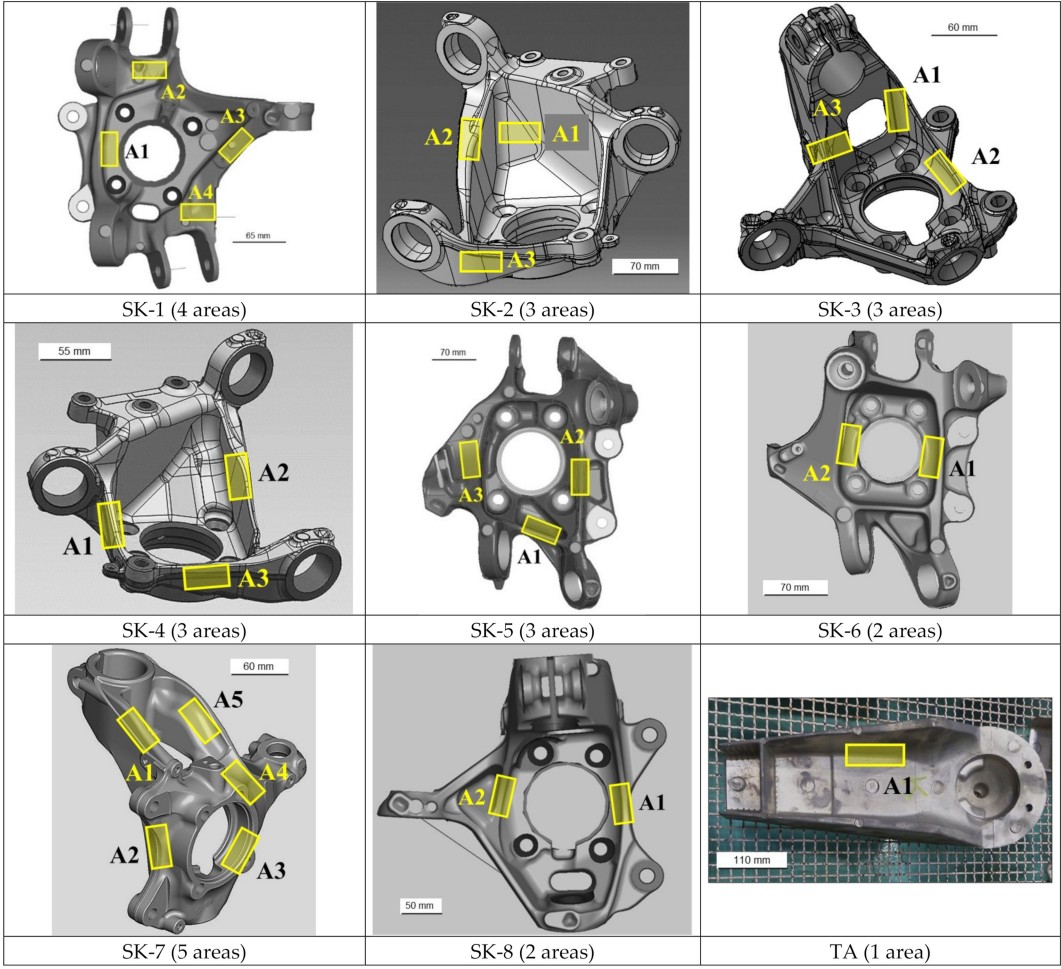

**Figure 6.** Different characterization areas selected in the industrial castings. The different areas studied in each casting are labeled as A1 to A5.

All values of the thermal parameters indicated in Figure 7, the $t_{SL}$, CR and $CR_{635}$ values obtained from them, and the experimentally measured SDAS values are listed in Table 6. The evolution of experimental SDAS with $t_{SL}$ and with CR has been plotted in the two graphs shown in Figure 8. Additionally, the SDAS values calculated by using Equations (2) (solidification time) and (3) (cooling rate) and the expressions reported by Chen et al. [18] have been also plotted in Figure 8 for comparative analysis.

### 3.2.1. Determination of SDAS

In case of the industrial castings, recording experimental cooling curves by inserting thermocouples at different areas was not possible as production dies were used. Hence simulated cooling curves have been alternatively used to firstly determine the four thermal parameters following analogous criteria to those indicated in Figure 4 and then to obtain the $t_{SL}$ and CR values in the selected areas of the SK castings. Examples of the simulated curves obtained for areas A2 and A4 of casting SK-1 with different cooling rates are given in Figure 7 where $T_{Al}$, $t_{Al}$, $T_{Eutectic}$ and $t_{Eutectic}$ parameters are indicated.

It is observed that all calculated curves for both $t_{SL}$ (Figure 8a) and CR (Figure 8b) parameters overestimate the experimental values of SDAS, and this point is especially relevant in case of the two equations reported by Chen et al. [18]. The different locations of the curves obtained from Equations (2) and (3), and from Chen et al. with respect to the experimental SDAS values are due to the differences found in coefficients K and n in each equation used to determine this structural parameter. As already mentioned before, it is reported that these two coefficients change according to the alloy composition, the casting

geometry, and the cooling conditions [18,21]. This comparative study involves a group of cast aluminum alloys (castings SK-1 to SK-6) with similar compositions, so the different casting variables and cooling kinetics used in each set of data should be the causes of the behaviors shown in Figure 8. Thus, the calculation of SDAS for the industrial castings manufactured by LPDC in this work is made using Equations (5) and (6) which are the best least square fit of the experimental SDAS data listed in Table 6 and the corresponding $t_{SL}$ and CR data (solid lines in Figure 8a,b).

$$\text{SDAS (μm)} = 6.331 \cdot (t_{SL})^{0.362} \quad R^2 = 0.924 \tag{5}$$

$$\text{SDAS (μm)} = 25.639 \cdot (CR)^{-0.351} \quad R^2 = 0.922 \tag{6}$$

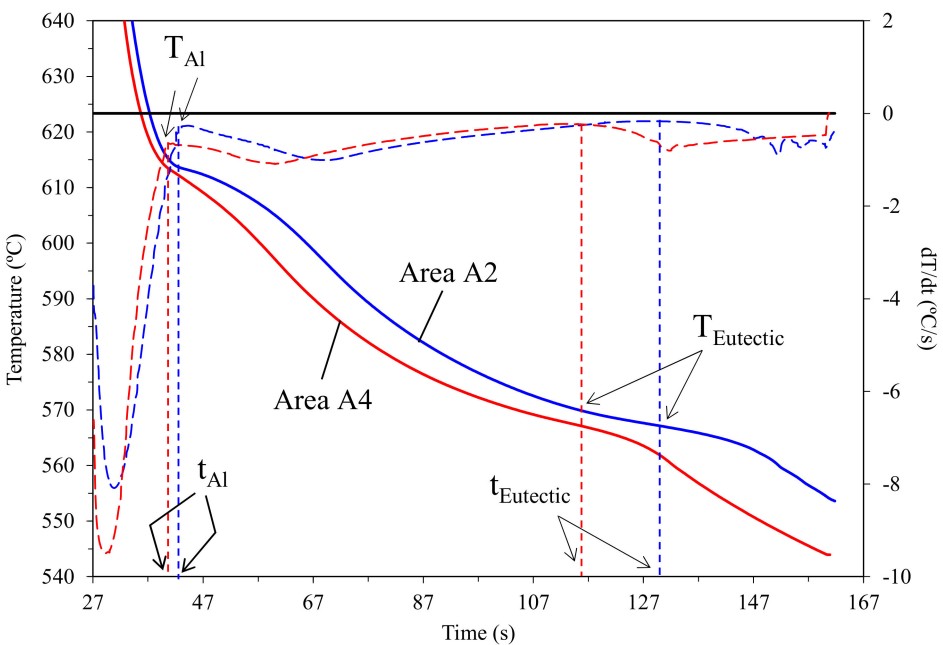

**Figure 7.** Characteristic solidification parameters in the simulated cooling curves (solid lines) obtained from areas A2 and A4 in casting SK-1. First derivative curves are plotted as interrupted lines.

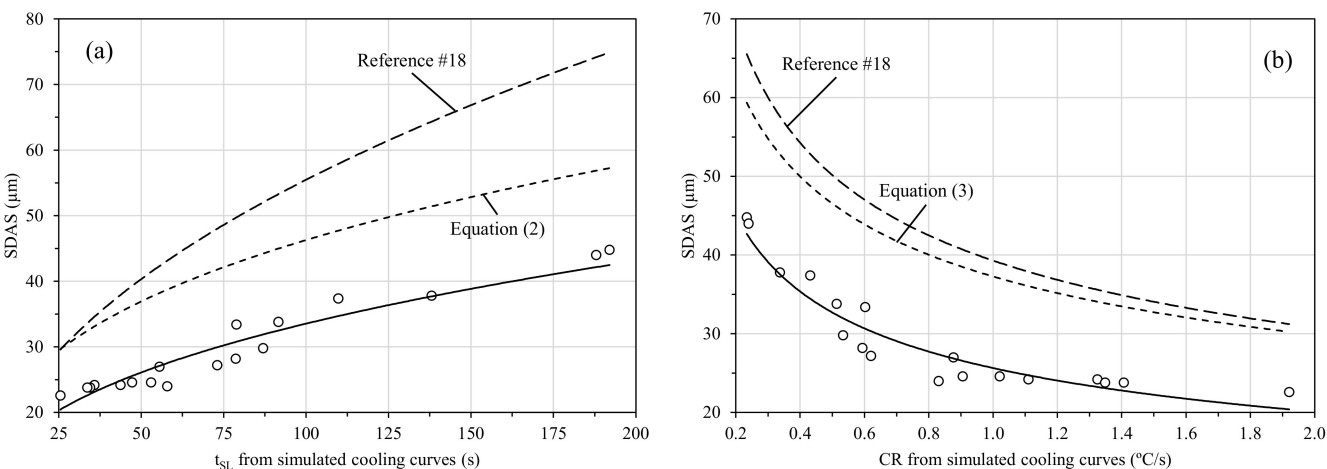

**Figure 8.** Evolution of experimental (dots and solid lines) and calculated SDAS (interrupted and dotted lines) in the SK-1 to SK-6 castings with (**a**) $t_{SL}$ and (**b**) CR, both parameters determined from the simulated cooling curves. Interrupted lines correspond to data from [18].

**Table 6.** Individual values of the solidification parameters determined from the simulated cooling curves obtained in each area of castings SK-1 to SK-6, of $t_{SL}$, CR and $CR_{635}$ and of the experimentally measured SDAS (standard deviation values are shown between brackets for this last parameter).

| Casting | Area | $T_{Al}$ (°C) | $t_{Al}$ (s) | $T_{Eutectic}$ (°C) | $t_{Eutectic}$ (s) | $t_{SL}$ (s) | CR (°C/s) | $CR_{635}$ (°C/s) | SDAS (μm) |
|---|---|---|---|---|---|---|---|---|---|
| SK-1 | A1 | 613.7 | 46.7 | 567.2 | 184.8 | 138.1 | 0.34 | 4.7 | 37.8 (1.2) |
| | A2 | 613.5 | 43.1 | 567.1 | 130.1 | 87.0 | 0.53 | 8.1 | 29.8 (1.0) |
| | A3 | 613.5 | 40.1 | 566.5 | 132.6 | 91.6 | 0.51 | 6.6 | 33.8 (1.7) |
| | A4 | 612.5 | 42.1 | 567.2 | 115.1 | 73.0 | 0.62 | 9.5 | 27.2 (1.3) |
| SK-2 | A1 | 613.6 | 24.9 | 566.3 | 134.7 | 109.8 | 0.43 | 5.9 | 37.4 (0.5) |
| | A2 | 612.8 | 16.8 | 564.7 | 74.7 | 57.9 | 0.83 | 14.9 | 24.0 (0.9) |
| | A3 | 613.4 | 24.9 | 566.7 | 106.3 | 78.7 | 0.59 | 8.5 | 28.2 (1.0) |
| SK-3 | A1 | 611.7 | 15.9 | 564.3 | 51.7 | 35.8 | 1.32 | 23.9 | 24.2 (0.7) |
| | A2 | 613.6 | 14.0 | 564.7 | 39.5 | 25.5 | 1.92 | 25.8 | 22.6 (1.4) |
| | A3 | 612.9 | 21.9 | 565.4 | 100.8 | 78.9 | 0.60 | 9.8 | 33.4 (1.0) |
| SK-4 | A1 | 612.3 | 21.1 | 564.3 | 74.1 | 53.0 | 0.91 | 19.2 | 24.6 (0.8) |
| | A2 | 611.9 | 19.9 | 565.4 | 54.4 | 34.5 | 1.35 | 27.0 | 23.8 (0.7) |
| | A3 | 612.7 | 19.9 | 564.2 | 63.6 | 43.7 | 1.11 | 20.9 | 24.2 (1.6) |
| SK-5 | A1 | 612.6 | 15.9 | 565.3 | 49.6 | 33.6 | 1.41 | 25.3 | 23.8 (1.0) |
| | A2 | 613.0 | 17.3 | 564.8 | 64.6 | 47.2 | 1.02 | 17.5 | 24.6 (0.5) |
| | A3 | 613.0 | 17.0 | 564.3 | 72.5 | 55.5 | 0.88 | 19.4 | 27.0 (0.6) |
| SK-6 | A1 | 613.1 | 25.3 | 568.2 | 217.3 | 192.0 | 0.23 | 3.2 | 44.8 (0.7) |
| | A2 | 613.4 | 27.7 | 568.5 | 215.7 | 188.0 | 0.24 | 3.2 | 44.0 (0.9) |

As it was found for the cylindrical test castings, another correlation with interest is observed here between the experimental SDAS and the $CR_{635}$ values obtained from the simulated cooling curves (Table 6). Data from these two parameters have been plotted in Figure 9, together with the curve calculated with Equation (4). In case of the industrial castings, Equation (4) predicts SDAS values a bit higher than the experimental ones, these last with similar scattering than those found for SDAS data plotted against $t_{SL}$ and CR (see Figure 8). The best least square fit of the experimental data plotted in Figure 9 leads to Equation (7):

$$\text{SDAS (μm)} = 59.638 \cdot (CR_{635})^{-0.296} \quad R^2 = 0.922 \tag{7}$$

Equations (5)–(7) already obtained to determine SDAS from a given simulated cooling curve have been used now to predict this structural parameter in the five areas of casting SK-7 and in the two areas of casting SK-8 indicated in Figure 6. Table 7 lists the solidification parameters obtained from the seven cooling curves simulated in these locations. Both experimental and predicted SDAS values, these last determined using the values listed in Table 7, are finally compared in Table 8. Experimental SDAS data given in this last table correspond to the average and the standard deviation values obtained from the set of five castings SK-7 and SK-8.

All three equations used in Table 8 lead to predicted SDAS values with differences with respect to the experimental ones which are similar to the range defined by ±2 σ, where σ is the standard deviation of the experimental data. The most relevant deviations are found for the highest SDAS experimental value in area A5 of casting SK-7 for those predictions made with Equations (5) and (6) though these differences have not been considered important enough to reject any of them.

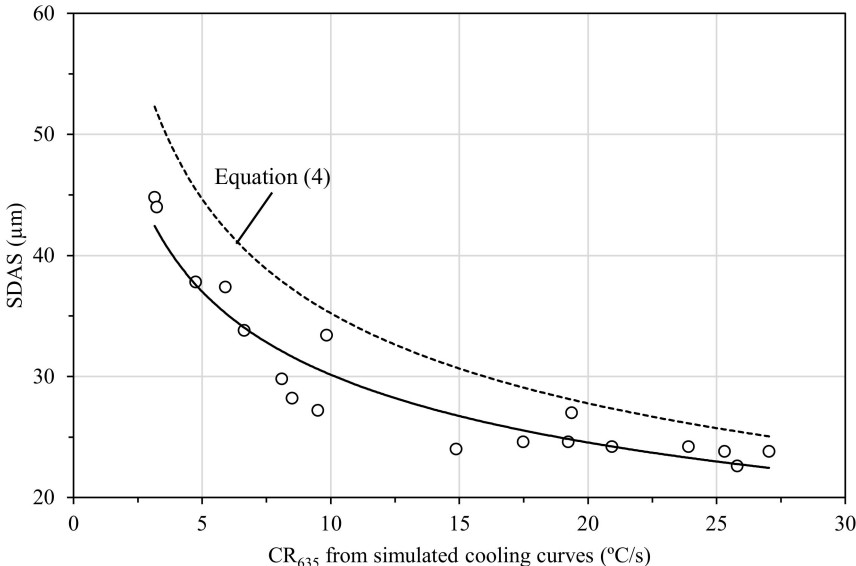

**Figure 9.** Evolution of experimental and calculated SDAS in the SK-1 to SK-6 castings with $CR_{635}$ parameter.

**Table 7.** Individual values of the solidification parameters determined from the simulated cooling curves obtained in the areas of castings SK-7 and SK-8.

| Casting | Area | $T_{Al}$ (°C) | $t_{Al}$ (s) | $T_{Eutectic}$ (°C) | $t_{Eutectic}$ (s) | $t_{SL}$ (s) | CR (°C/s) | $CR_{635}$ (°C/s) |
|---------|------|---------------|--------------|---------------------|--------------------|--------------|-----------|-------------------|
| SK-7 | A1 | 612.5 | 23.4 | 564.4 | 64.9 | 41.6 | 1.16 | 25.6 |
| | A2 | 613.0 | 25.1 | 565.6 | 80.7 | 55.6 | 0.85 | 14.4 |
| | A3 | 612.6 | 23.4 | 566.6 | 52.7 | 29.3 | 1.57 | 19.2 |
| | A4 | 613.5 | 34.1 | 566.5 | 152.7 | 118.6 | 0.40 | 6.3 |
| | A5 | 613.7 | 34.1 | 568.4 | 130.0 | 95.9 | 0.47 | 5.6 |
| SK-8 | A1 | 613.4 | 27.7 | 567.1 | 83.5 | 55.8 | 0.83 | 15.7 |
| | A2 | 613.0 | 20.9 | 567.5 | 76.3 | 55.4 | 0.82 | 17.6 |

**Table 8.** Experimental and calculated SDAS values in the areas of castings SK-7 and SK-8 (standard deviation values are shown between brackets).

| Casting | Area | SDAS (μm) | | | |
|---------|------|-----------|--------------|--------------|--------------|
| | | Experimental | Equation (5) | Equation (6) | Equation (7) |
| SK-7 | A1 | 24.8 (1.3) | 24.4 | 24.4 | 22.8 |
| | A2 | 25.4 (0.8) | 27.1 | 27.1 | 27.1 |
| | A3 | 23.8 (0.7) | 21.5 | 21.9 | 24.9 |
| | A4 | 33.2 (1.0) | 35.7 | 35.4 | 34.6 |
| | A5 | 37.2 (1.2) | 33.0 | 33.4 | 35.8 |
| SK-8 | A1 | 27.2 (1.0) | 27.1 | 27.4 | 26.9 |
| | A2 | 26.2 (1.4) | 27.1 | 27.5 | 26.0 |

### 3.2.2. Determination of Quality Index

The quality index (QI) [22] is frequently used to assess both strength and ductility performance of A356 and other cast aluminum alloys with different microstructural characteristics [23,24], though a high value of this parameter has also been related to the fine distribution

of dendrites, low microporosities [25], and absence of metallurgical defects [26–28]. In this context, QI can be determined according to Equation (8) [25,29]:

$$QI = UTS + 150 \cdot \log(A) \qquad (8)$$

Determination of tensile properties on industrial castings is usually defined by requirements from customers. However, determination of these properties in some small sections considered essentials for guaranteeing the correct functionality of the casting can be complex due to geometric issues. In this sense, it is worthy to obtain here an expression for determining QI in each of the selected areas by using the SDAS values experimentally measured on these areas or using those calculated with the equations obtained in the present work.

Tensile results from each area of castings SK-1 to SK-6 and the corresponding QI values determined according to Equation (8) are listed in Table 9. Data from this last parameter have been then plotted against the corresponding experimental SDAS values leading to the trend shown in Figure 10. Additionally, two series of SDAS and QI data reported by Khomamizadeh et al. [25] have been also plotted in Figure 10 for comparing the results obtained in the present work with A356 alloys cast in sand molds with comparatively low cooling rates.

**Table 9.** Average values of UTS, YS, A and QI for each area analyzed in castings SK-1 to SK-6 (standard deviation values are shown between brackets).

| Casting | Area | UTS (MPa) | YS (MPa) | A (%) | QI (MPa) |
|---|---|---|---|---|---|
| SK-1 | A1 | 297.8 (2.1) | 231.4 (2.9) | 6.9 (0.3) | 424.0 |
| | A2 | 308.0 (2.0) | 241.3 (1.6) | 8.9 (0.8) | 450.3 |
| | A3 | 299.6 (2.3) | 237.4 (2.9) | 8.0 (0.2) | 434.7 |
| | A4 | 313.6 (3.9) | 245.1 (4.3) | 10.0 (1.9) | 463.3 |
| SK-2 | A1 | 295.0 (3.0) | 221.4 (3.1) | 9.5 (1.3) | 441.5 |
| | A2 | 303.8 (3.5) | 228.2 (2.3) | 14.3 (0.9) | 477.2 |
| | A3 | 301.4 (2.7) | 225.4 (4.1) | 11.4 (0.8) | 460.2 |
| SK-3 | A1 | 304.2 (1.7) | 230.0 (1.1) | 14.1 (1.4) | 476.8 |
| | A2 | 307.8 (4.4) | 232.4 (3.6) | 14.0 (2.7) | 479.8 |
| | A3 | 298.0 (1.7) | 220.9 (2.7) | 9.4 (0.8) | 443.7 |
| SK-4 | A1 | 308.4 (2.3) | 231.6 (2.5) | 12.3 (2.0) | 474.7 |
| | A2 | 304.9 (2.1) | 226.4 (2.6) | 12.9 (0.5) | 471.3 |
| | A3 | 308.1 (4.5) | 227.4 (4.9) | 14.6 (1.7) | 482.9 |
| SK-5 | A1 | 318.4 (2.5) | 232.5 (2.1) | 13.0 (1.6) | 485.4 |
| | A2 | 313.6 (2.2) | 229.6 (2.6) | 13.2 (1.8) | 481.7 |
| | A3 | 308.6 (2.7) | 224.4 (2.4) | 13.3 (1.7) | 477.3 |
| SK-6 | A1 | 290.8 (2.5) | 215.6 (7.2) | 8.4 (0.9) | 429.1 |
| | A2 | 291.8 (4.7) | 213.6 (4.1) | 7.6 (0.1) | 424.3 |

Khomamizadeh et al. [25] used an experimental layout with a copper chill piece stuck to the smallest side of their wedges cast in silicate bonded sand molds. Thus, they could increase the cooling rate of the alloy from this chilled side of the casting to the opposite one where a riser was set. Although no data about the weight of the produced wedges is reported, their sections are larger than those present in all castings studied in the present work. Also, these authors used two different copper chill sizes and thus originated two cooling series namely A and C (see Figure 10 caption). As it can be seen in Figure 10, QI increases when decreasing SDAS in all plotted series, though it has observed that this

a correlation seems to achieve a maximum value at about 480–490 MPa from the data obtained in the present work. This maximum range is defined by the maximum cooling rate achieved in the present investigation. The best linear least square fit of the series of data plotted in the small graph inserted in Figure 10 leads to Equation (9) which will be used in Section 3.2.6 for checking the QI values in the different areas studied in castings SK-7 and SK-8.

$$[QI\ (MPa)] = 544.460 - 2.850 \cdot SDAS\ R^2 = 0.876 \tag{9}$$

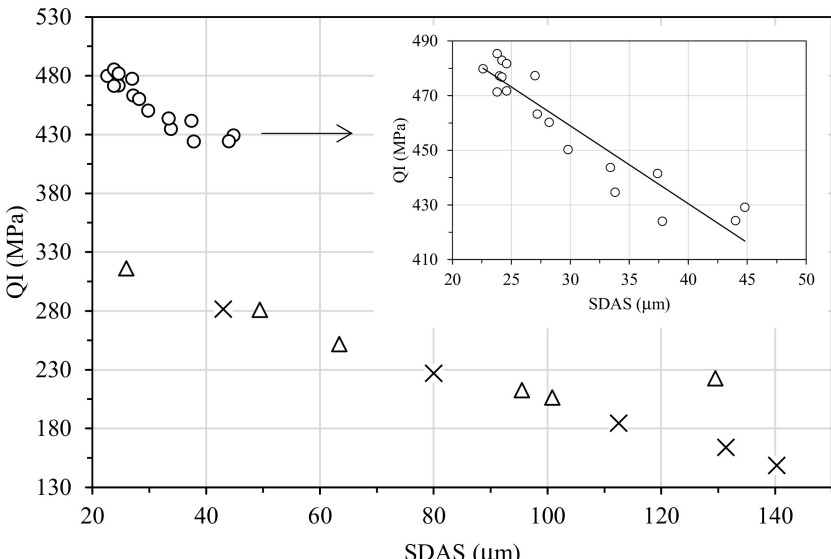

**Figure 10.** Evolution of experimental SDAS with QI in castings SK-1 to SK-6 (open circles), and in series A (crosses) and series C (open triangles) from Khomamizadeh et al. [25]. The small graph inside shows an extended view of data obtained in the present work with the best linear least square fit of these data.

When analyzing the data from reference [25], the effect of using apparently big wedges cast in sand molds led to cooling rates lower than those occurred in the SK and TA castings produced in a LPDC facility. Thus, SDAS and QI values are also lower in the former cases than in the later ones as it is seen in Figure 10.

### 3.2.3. Effect of SDAS on Tensile Properties

Tensile properties of AlSiMg cast alloys mainly depend on their composition, being Mg content the most important parameter, and on microstructural characteristics [20,21,30,31]. These last features are defined by the solidification path of the cast alloy (defined by casting processes) and by the parameters used in the heat-treatments (HT) performed [32]. In this section, experimental SDAS values and tensile properties measured in each area of castings SK-1 to SK-6 have been correlated to study the role of this structural parameter under two HT conditions. In subsequent sections, the effect of Mg contents and different HT parameters on tensile properties will be dealt and a final model will be developed with the obtained results to predict these mechanical properties.

The graphs included in Figure 11 shows the evolutions of average values of UTS, YS, and A with experimentally measured SDAS values for castings SK-1 in one hand, and for castings SK-2 to SK-6 on the other hand. As mentioned in Section 3.2.2, this distinction is made as the aging conditions were different in these two cases (see Table 2). It is observed from the graphs shown in Figure 11 that all three tensile parameters decrease when increasing SDAS for the two HT conditions used. The correlations found have been considered linear though an apparent lack of any effect from SDAS is detected at values lower than 25 μm when the aging temperature and time were 155 °C and 360 min, respectively. This aspect is mainly observed in case of UTS (see Figure 11a). After analyzing

the data from each casting in this series, it is found that this lack of effect is likely due to two characteristics: on one hand, SK-5 casting was produced with the highest Mg content (0.38 wt.%) among the SK-1 to SK-6 group and this fact increases UTS and YS data (see the three arrows in Figure 11a for the former property). On the other hand, some scattering of UTS, YS and A is present in this SDAS range as it contains the highest amount of data.

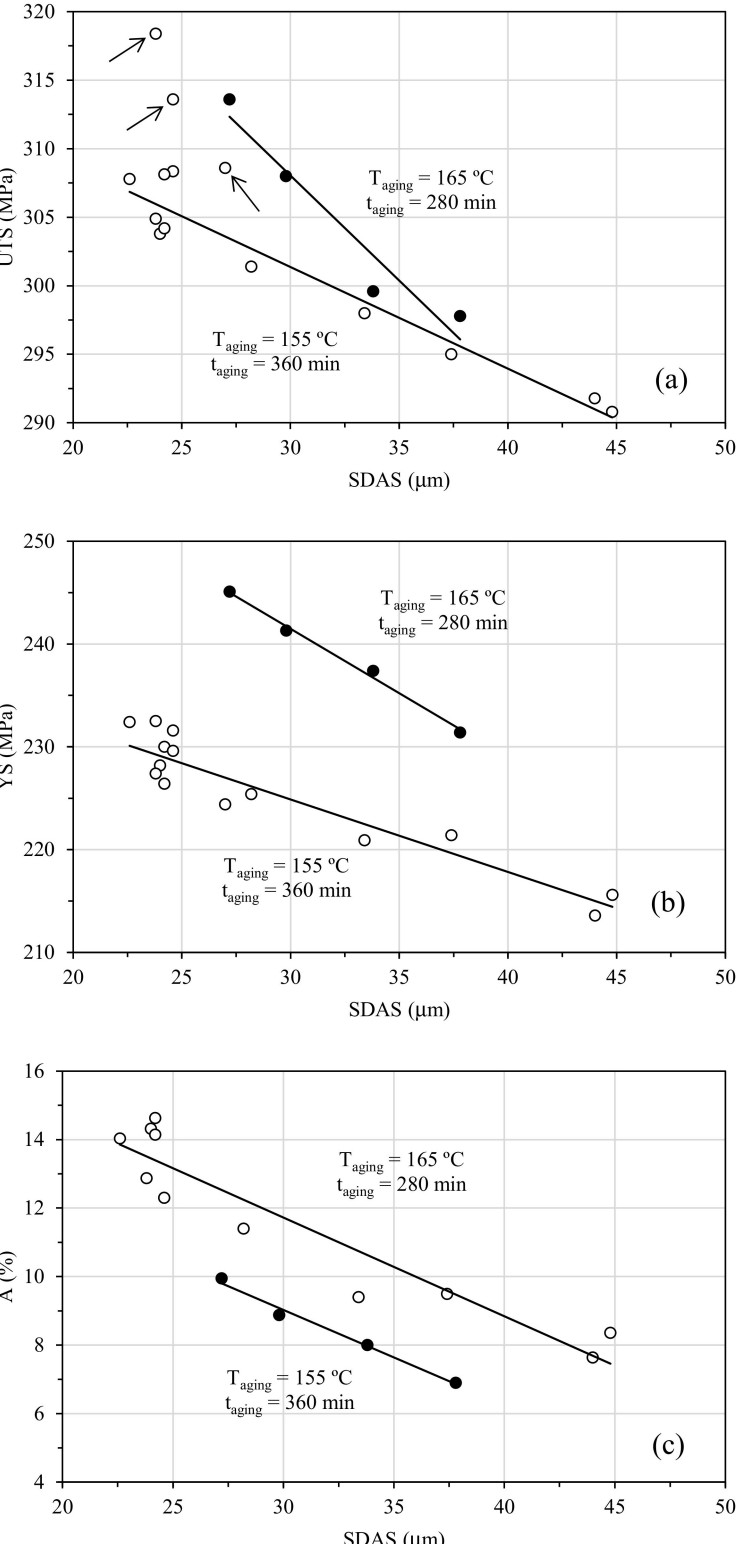

**Figure 11.** Evolution of UTS (**a**), YS (**b**) and A (**c**) with the experimental SDAS in castings SK-1 (solid circles) and in castings SK-2 to SK-6 (open circles).

The best linear least square fits of each evolution are also plotted in the graphs shown in Figure 11. In case of UTS, the three values pointed with arrows in Figure 11a were not considered in the corresponding fit because of the high Mg content which will be studied in Section 3.2.5. Thus, the following expressions are obtained:

$$[\text{UTS (MPa)}]_{165°C/280min} = 354.071 - 1.534 \cdot \text{SDAS} \quad R^2 = 0.930 \tag{10}$$

$$[\text{YS (MPa)}]_{165°C/280min} = 279.158 - 1.255 \cdot \text{SDAS} \quad R^2 = 0.993 \tag{11}$$

$$[\text{A (\%)}]_{165°C/280min} = 17.349 - 0.277 \cdot \text{SDAS} \quad R^2 = 0.987 \tag{12}$$

$$[\text{UTS (MPa)}]_{155°C/360min} = 323.613 - 0.741 \cdot \text{SDAS} \quad R^2 = 0.935 \tag{13}$$

$$[\text{YS (MPa)}]_{155°C/360min} = 246.071 - 0.706 \cdot \text{SDAS} \quad R^2 = 0.910 \tag{14}$$

$$[\text{A (\%)}]_{155°C/360min} = 20.310 - 0.286 \cdot \text{SDAS} \quad R^2 = 0.883 \tag{15}$$

Another relevant aspect in Figure 11a is the intersection of the two linear fits at about SDAS = 38 μm. Above this value castings aged at 165 °C for 280 min would not be effective for obtaining UTS values higher than those achieved with the low temperature aging treatment. Also notice that such an intersection for YS (Figure 11b) could be found at SDAS values higher than 55 μm while almost parallel lines are obtained for elongation.

Figure 12 shows the cross-sections of the fracture surfaces found in two tensile specimens obtained from areas A1 and A4 of casting SK-1 with SDAS values of 37.8 μm and 27.2 μm, respectively. Notice that tearing follows the interdendritic areas in both cases though they become the biggest for the sample with the highest SDAS value (Figure 12a,b). This fact leads to decreases in all three UTS, YS and A parameters as observed in the graphs shown in Figure 11.

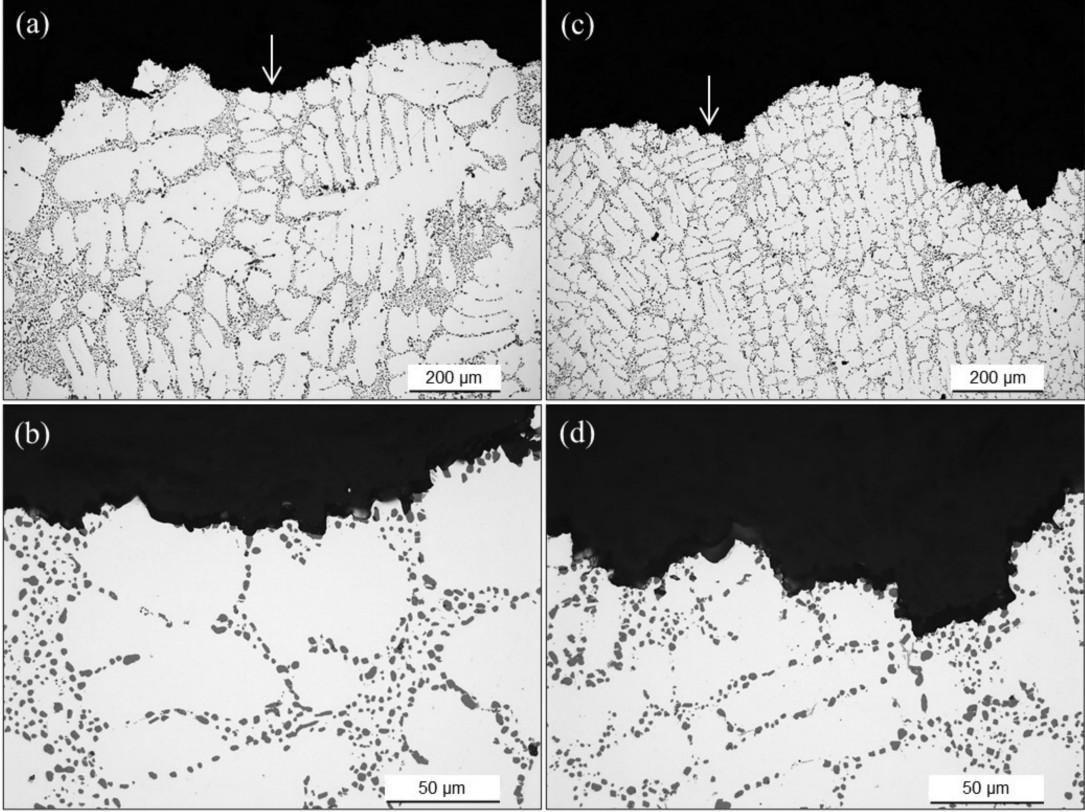

**Figure 12.** Cross sections of the tensile specimens obtained from casting SK-1: (**a**,**b**) area A1 with SDAS of 37.8 μm, (**c**,**d**) area A4 with SDAS of 27.2 μm. Images (**b**,**d**) correspond to high magnification views of the zones indicated with white arrows in images (**a**,**c**), respectively.

### 3.2.4. Effect of the Aging Phase of T6 on Tensile Properties and Microstructure

An important step for developing the predictive model is to study the influence of HT, namely the aging parameters, on tensile properties. As mentioned in Section 2, TA castings were exclusively used in this investigation for this purpose with a unique area of study as shown in Figure 6. Thus, groups of 12 castings were heat-treated following each aging condition listed in Table 3. All the SDAS values measured in the A1 area of these castings are in the range of 27.7–29.0 μm. Table 10 lists all tensile results obtained in this part of the work.

The graphs shown in Figure 13 show the evolution of UTS, YS and A with aging time for the two different aging temperatures used in this study. These graphs also show the best least square fits for each aging temperature following logarithmic equations. Both ultimate tensile strength and yield strength increase with $t_{aging}$ for a given temperature showing a saturation effect at the highest $t_{aging}$ values. Once achieved a given time, most of the fine Mg-Si compounds are already precipitated as $\beta$-$Mg_2Si$ in the aluminum matrix [33,34] and both UTS and YS show increases beyond that time value due to this coarsening of Mg-Si particles. In this sense, a minimum aging time of 250–300 min seems necessary so as to obtain relevant values of these two tensile strength parameters while keeping A values above 11%.

**Table 10.** Average values of UTS, YS and A for each aging temperature and time in castings TA (standard deviation values are shown between brackets).

| $T_{aging}$ (°C) | $t_{agging}$ (min) | UTS (MPa) | YS (MPa) | A (%) |
|---|---|---|---|---|
| 165 | 50 | 257.5 (1.7) | 137.8 (1.1) | 18.4 (1.2) |
| | 100 | 271.5 (5.0) | 160.0 (5.5) | 16.1 (0.8) |
| | 200 | 292.0 (6.5) | 214.5 (5.2) | 12.6 (0.6) |
| | 280 | 314.3 (4.7) | 234.1 (6.4) | 10.7 (1.4) |
| | 400 | 325.3 (5.8) | 256.6 (5.2) | 9.5 (1.7) |
| | 460 | 327.9 (3.3) | 263.5 (2.0) | 9.3 (1.5) |
| 155 | 50 | 253.8 (2.8) | 132.5 (2.7) | 17.3 (1.2) |
| | 100 | 257.3 (3.1) | 139.8 (2.5) | 17.1 (2.1) |
| | 200 | 280.8 (4.7) | 180.5 (4.6) | 12.6 (2.0) |
| | 360 | 307.8 (5.5) | 226.5 (5.7) | 10.1 (1.7) |
| | 480 | 308.2 (6.5) | 234.4 (9.0) | 9.0 (1.6) |
| | 510 | 312.0 (5.3) | 235.6 (3.2) | 9.8 (1.2) |
| | 600 | 313.8 (4.3) | 240.3 (3.5) | 8.7 (1.5) |

For elongation at rupture, increasing $t_{aging}$ leads to reductions of this parameter also showing a saturation effect at the highest aging times. The hardening effect due to precipitation of small Mg-Si particles originates an opposite evolution of A when compared to those observed from UTS and YS. This result contrasts with the effect of SDAS on tensile properties (see the graphs in Figure 11) as a refinement of microstructure during solidification leads to an improvement of homogeneity and all three UTS, YS and A parameters are increased.

The positive effect of the highest aging temperature on UTS and YS evolutions is seen in Figure 13a,b where 250–325 min are needed at $T_{aging}$ = 165 °C to obtain similar strengths to the maximum values achieved at $T_{aging}$ = 155 °C after 475–600 min. Furthermore, longer times in case of the highest temperature led to UTS and YS values higher than 325 MPa and 260 MPa, respectively. Interestingly Figure 13c shows that the two aging temperatures used in the present work barely affects to elongation at rupture showing that this parameter is quite sensitive to precipitation of Mg-Si compounds.

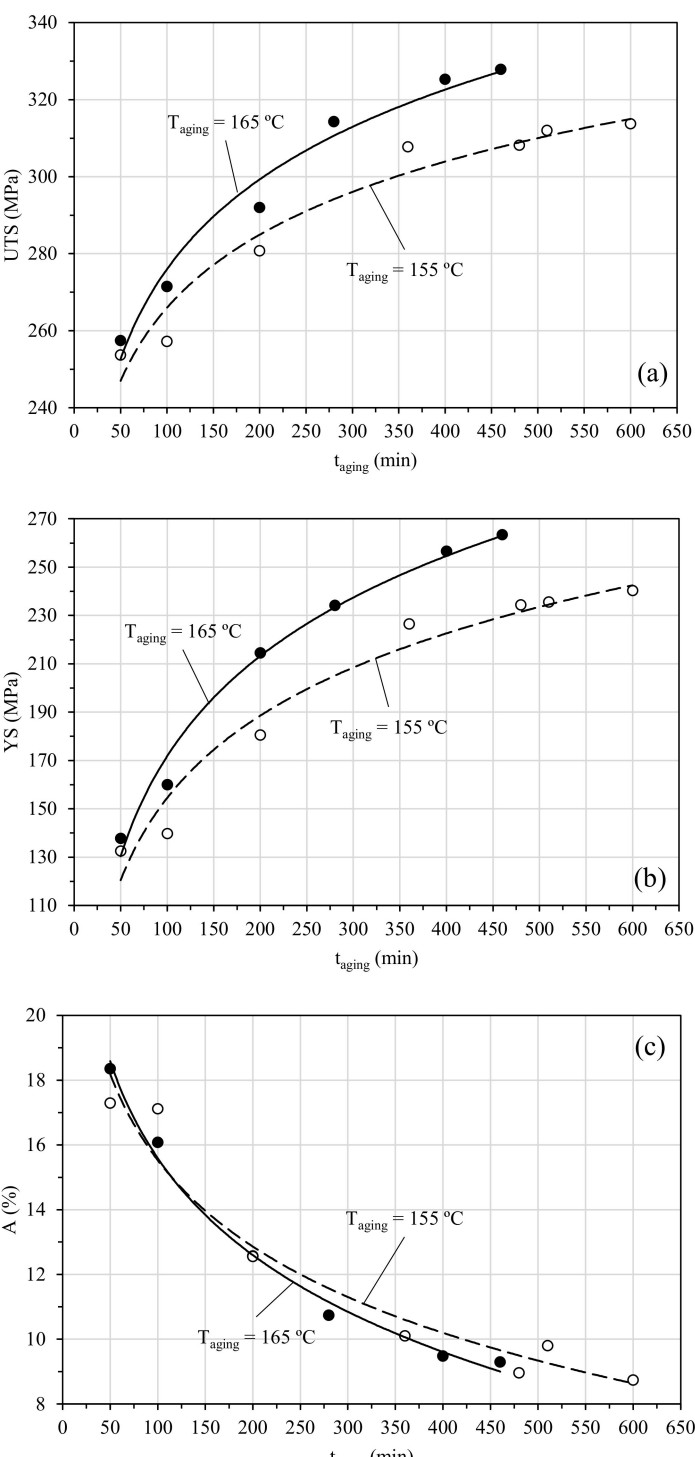

**Figure 13.** Evolution of UTS (**a**), YS (**b**) and A (**c**) with aging time in castings TA at 165 °C (solid circles) and at 155 °C (open circles). Best least square fits are plotted as a solid line and as an interrupted line for $T_{aging}$ = 165 °C and 155 °C, respectively.

As already said above all best least square fits shown in Figure 13 were based on logarithmic curves which lead to the following expressions:

$$[\text{UTS (MPa)}]_{165°C} = 33.624 \cdot \ln(t_{aging}) + 121.141 \quad R^2 = 0.973 \tag{16}$$

$$[\text{YS (MPa)}]_{165°C} = 59.746 \cdot \ln(t_{aging}) - 103.322 \quad R^2 = 0.985 \tag{17}$$

$$[A\ (\%)]_{165°C} = -4.315 \cdot \ln(t_{aging}) + 35.460 \quad R^2 = 0.992 \tag{18}$$

$$[UTS\ (MPa)]_{155°C} = 27.361 \cdot \ln(t_{aging}) + 139.977 \quad R^2 = 0.955 \tag{19}$$

$$[YS\ (MPa)]_{155°C} = 49.111 \cdot \ln(t_{aging}) - 71.643 \quad R^2 = 0.961 \tag{20}$$

$$[A\ (\%)]_{155°C} = -3.844 \cdot \ln(t_{aging}) + 33.230 \quad R^2 = 0.949 \tag{21}$$

Figure 14 shows the microstructure changes observed on the analyzed area of TA castings due to the solubilization and quenching steps. Before undergoing the T6 treatment, interdendritic areas with comparatively fine Si particles and some coarse dark $Mg_2Si$ particles are detected (some of them are included in circles pointed with red arrows in Figure 14b). After the T6 treatment the Si particles changed to coarse and rounded ones which appear distributed in these interdendritic areas (Figure 14c,d). This last type of microstructure remains present in all castings then aged at different temperatures and times as it is illustrated in Figure 14e,f for a TA casting aged at 155 °C for 200 min. Hence the increases of UTS and YS with $T_{aging}$ and/or $t_{aging}$ and the decrease of A when increasing these two aging parameters (Figure 13) correspond to $Mg_2Si$ precipitates formed in the $\alpha$-Al dendrites which are not detectable by optical metallographic analyses [33].

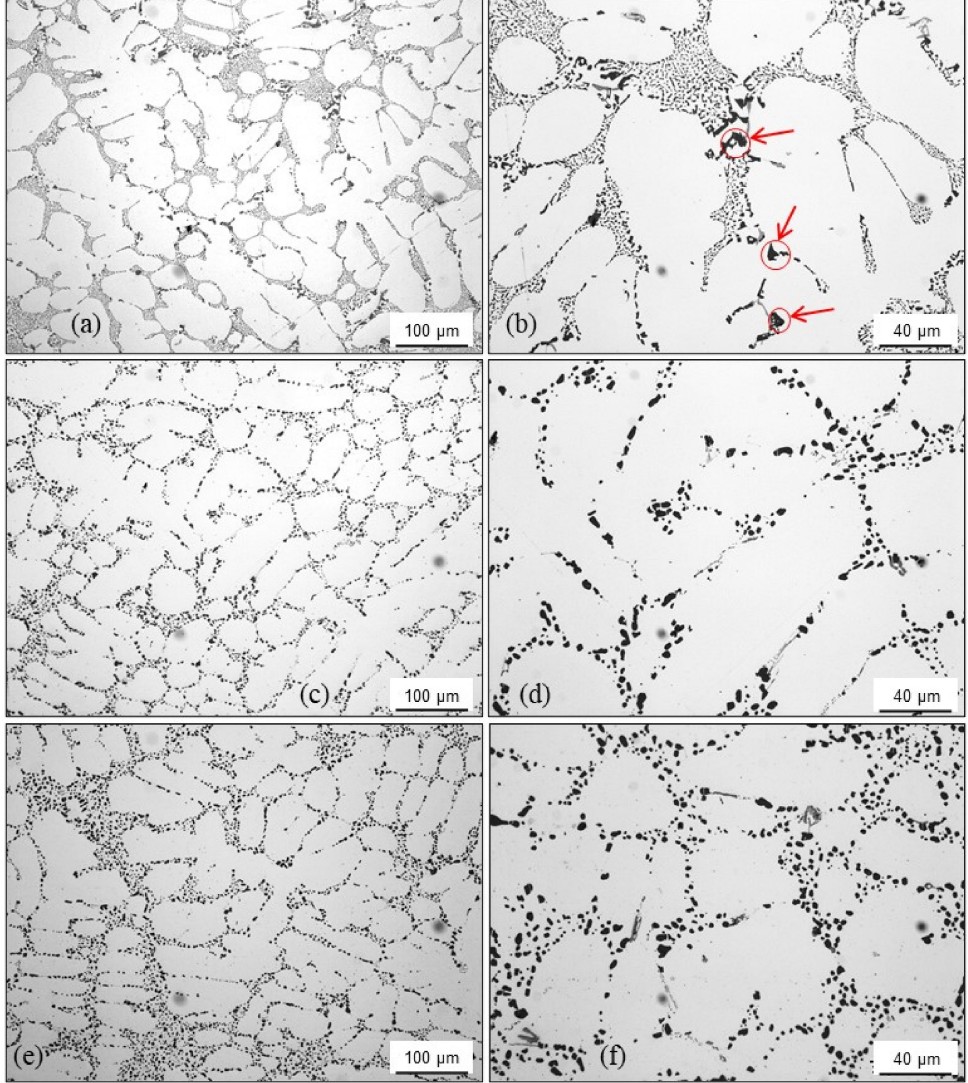

**Figure 14.** Optical micrographs of the alloy A356.2 used to produce the TA castings: (**a**,**b**) as cast microstructure, (**c**,**d**) after solubilization and quenching steps. Images (**e**,**f**) show the microstructure observed in a TA casting aged at 155 °C for 200 min.

3.2.5. Effect of Mg Content on Tensile Properties

As it has been already said in previous sections, one of the most relevant parameter with influence on mechanical properties of A356 alloys is the Mg content present in the matrix after solution treatment and quenching steps [33–35]. This element is then intentionally precipitated as $Mg_2Si$ small-sized compounds via subsequent artificial aging treatments like those described in Table 2. The industrial practice focused by the present investigation commonly uses Mg contents which range from 0.30 wt.% to 0.45 wt.% and castings SK-1 to SK-6 and TA were manufactured into this range (see Table 2). However, only the TA casting was produced using two batches with apparently close Mg contents, namely 0.35 wt.% and 0.38 wt.%, which originated a short range for this alloying element.

Thus, two sets of data from Möller et al. [36,37], and another one from Fortini et al. [38], have been also analyzed in the present work to determine the effect of Mg content on tensile properties of A356 alloys and to compare them with data obtained from castings produced in the present work. Although Möller et al. [36,37] used semi-solid metal (SSM) slurries directly prepared from the liquid alloy to produce 4 mm × 80 mm × 100 mm plates by HPDC facility, they also reported similar artificial aging T6 responses for same automotive brake calipers manufactured using both SSM-A356 alloys with globular microstructures and A356 alloys in gravity die casting, this last leading to dendritic microstructures with SDAS values of about 20 μm [39]. In case of A356 alloys prepared by Fortini et al. [38], they were cast in a preheated die to produce castings with dendritic microstructures. Although these authors do not report SDAS values in their work, this parameter has been estimated into the range of 20–30 μm according to the images shown in their Figure 4.

Figure 15 shows the evolutions of tensile strength and yield strength with Mg content for SSM-A356 alloys [36,37] and for A356 alloys used in gravity die casting [37]. It is observed that UTS increases with Mg content showing a linear correlation which is not affected by the different casting processes or the different aging conditions used in each set of data (180 °C and 60 min for Möller et al. [36,37] and 155 °C and 270 min for Fortini et al. [38]). For YS, this parameter also increases with Mg content though two different linear correlations are found: on one hand for the two sets of data from Möller et al. [36,37] and on the other hand for the set of data from Fortini et al. [38]. Thus, YS data from Fortini et al. originate a linear distribution with similar slope and about 35 MPa lower than that found for data from Möller et al. Although this change can be due to the different microstructures present in these two groups of A356 alloys, it is remarkable that only YS parameter showed such a difference.

The best linear least square fits obtained for UTS and YS vs. Mg content are given by Equations (22)–(24) where the term "Mg" represents the weight content of this element in the alloy. Notice that all available UTS data were used to obtain Equation (22) while YS data were separately analyzed to give Equations (23) and (24) for Möller et al. [36,37] and Fortini et al. [38], respectively, according to the two fits shown in Figure 15.

$$\text{UTS (MPa)} = 225.057{\cdot}\text{Mg} + 244.393 \ R^2 = 0.930 \tag{22}$$

$$[\text{YS (MPa)}] = 202.410{\cdot}\text{Mg} + 184.081 \ R^2 = 0.982 \tag{23}$$

$$[\text{YS (MPa)}] = 223.457{\cdot}\text{Mg} + 153.688 \ R2 = 0.970 \tag{24}$$

The UTS (circles) and YS (triangles) values obtained from the SK-1 to SK-6 and TA castings aged at 165 °C for 280 min (red solid symbols) and at 155 °C for 360 min (blue solid symbols) with SDAS values lower than 30 μm are also plotted in Figure 15. As expected, red symbols lead to higher UTS and YS values than blue ones for a given Mg content due to the high aging temperature used in the former cases. Although some scattering is present in the data from this work likely due to variability of SDAS values, both UTS and YS show trends in good agreement with the ones obtained from data reported by Möller et al. [36,37] and from Fortini et al. [38].

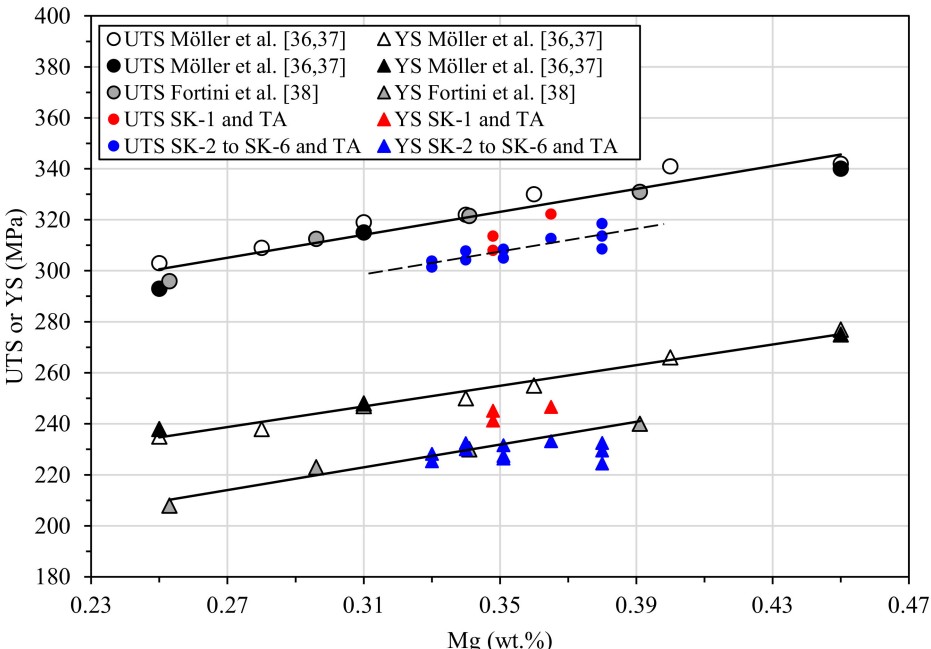

**Figure 15.** Effect of Mg content on UTS and YS for two series of SSM-A356 alloys from Möller et al. [36,37] and for one series of A356 alloys from Fortini et al. [38]. The different aging treatments were 180 °C and 60 min for alloys from Möller et al. while they were 155 °C and 270 min for alloys from Fortini et al. Red and blue symbols correspond to UTS (circles) and YS (triangles) values obtained from SK-1 to SK-6 and TA castings used in this work which were aged at 165 °C for 280 min and at 155 °C for 360 min, respectively (see the text).

Regarding elongation at rupture (Figure 16), the data from Fortini et al. show a decrease of this parameter when increasing the Mg content though the number of values reported is certainly low. In the series from Möller et al. Mg content does not show a relevant effect on their A values. However, it is observed that A increases in those alloys aged at comparatively low temperatures for a given Mg content. This is the case of the alloys reported by Fortini et al. which were aged at 155 °C with respect to those from Möller et al. aged at 180 °C for 60 min. This finding is confirmed by the results from the present work when comparing the A data obtained for alloys aged at 165 °C (red circles) with those from alloys aged at 155 °C (blue circles) in Figure 16.

Going back to the effect of Mg contents on elongation, the lack of a clear tendency in blue and red dots plotted in Figure 16 is probably related to the different areas studied in castings SK-1 to SK-6 and TA which were produced with a unique Mg content per area. Hence a satisfactory correlation between Mg content and A is not available with the data shown in Figure 16.

Thus it seems interesting to use here the elongation data from A1 and A2 areas of castings SK-7 and SK-8 with SDAS values in the range of 25–27 μm. These castings were produced with different Mg contents as it will be described in Section 3.2.6 (Table 12) and their elongation values are plotted in Figure 17 together with those reported by Fortini et al. [38]. It is found that A decreases when increasing Mg contents though the observed evolutions are slightly different according to the aging conditions used in each case. Assuming minor differences in SDAS among the four series of data shown in Figure 17, the highest aging temperature used (165 °C) sets the two series of data from SK-7 casting at the lowest elongation range. Comparing the other two series aged at 155 °C at the highest A values, it is found that higher aging times (data from SK-8 castings) increase the negative effect of increasing Mg contents on A property (more negative slope in the linear trend). This result is also observed when comparing the two series from castings SK-7 (both aged at 165 °C for 280 min and 360 min).

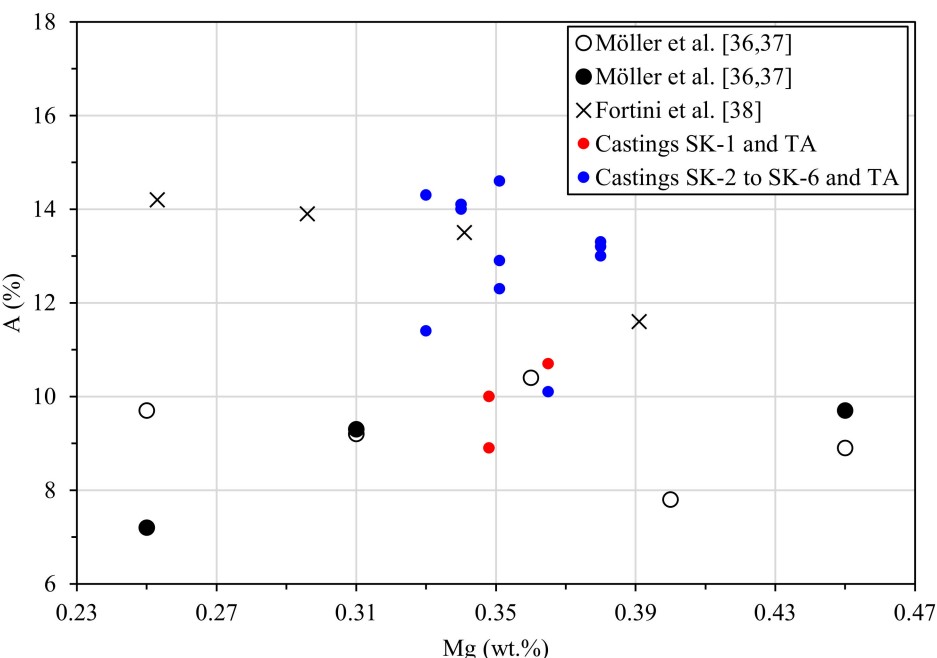

**Figure 16.** Evolution of A with Mg content for two series of SSM-A356 alloys from Möller et al. [36,37] and for one series of A356 alloys from Fortini et al. [38]. Red and blue symbols correspond to A values obtained from SK-1 to SK-6 and TA castings used in this work aged at 165 °C for 280 min and at 155 °C for 360 min, respectively (see the text).

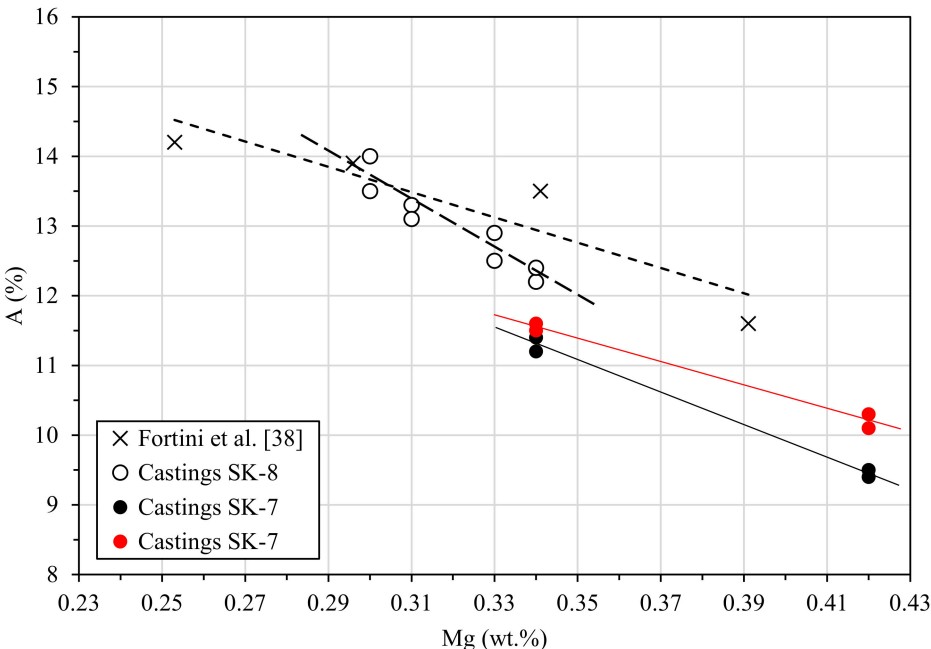

**Figure 17.** Evolution of A with Mg content for the series of A356 alloys reported by Fortini et al. [38] aged at 155 °C for 270 min (crosses), from areas A1 and A2 in castings SK-7 aged at 165 °C for 280 min (gray solid circles), for 360 min (black solid circles) and from areas A1 and A2 in castings SK-8 aged at 155 °C for 360 min (open circles).

The best linear least square fits of the four series of data plotted in Figure 17 lead Equations (25) and (26) where the slope coefficients and intercept are defined as a function of $t_{aging}$:

$$[A\ (\%)]_{165°C} = (-0.078 \cdot t_{aging} + 5.000) \cdot Mg + 0.023 \cdot t_{aging} + 10.725 \qquad (25)$$

$$[A \, (\%)]_{155°C} = (-0.176 \cdot t_{aging} + 29.485) \cdot Mg + 0.053 \cdot t_{aging} + 4.843 \tag{26}$$

### 3.2.6. Prediction Models and Validation

The dependence of SDAS values measured in different areas of SK castings on simulated cooling curves together with the effect of SDAS, aging treatments and Mg content on tensile properties has been separately studied in previous sections. In this part of the investigation, equations for predicting tensile properties are defined based on the results obtained in these previous studies as a series of terms which represent the contribution of each variable:

$$(P)_{T_{aging}} = P_{HT} + P_{SDAS} + P_{Mg} \tag{27}$$

where $(P)_{T_{aging}}$ is the predicted property for a given $T_{aging}$ value and $P_{HT}$, $P_{SDAS}$ and $P_{Mg}$ are the terms which represent the effect of HT, SDAS and Mg content on such a property, respectively. Following this schematic, equations from Sections 3.2.3–3.2.5 have been combined to obtain expressions to predict UTS, YS and A values for a given area of castings aged at 165 °C or 155 °C according to:

$$(UTS/YS)_{T_{aging}} = \alpha \cdot \ln(t_{aging}) + \beta + \gamma \cdot (SDAS - SDAS_0) + \delta_1 \cdot (Mg - Mg_0) \tag{28}$$

$$(A)_{T_{aging}} = \alpha \cdot \ln(t_{aging}) + \beta + \gamma \cdot (SDAS - SDAS_0) + (\delta_2 \cdot t_{aging} + \delta_3) \cdot (Mg - Mg_0) \tag{29}$$

where values for coefficients $\alpha$, $\beta$, $\gamma$, $\delta_1$, $\delta_2$ and $\delta_3$ are given in Table 11 for each property and aging temperature. Equation numbers used to obtain these coefficients are also indicated in this table. $SDAS_0$ and $Mg_0$ parameters stand for the average SDAS value of the TA castings and the average Mg content at which these castings were produced for studying the effect of aging parameters on tensile properties (Section 3.2.4).

**Table 11.** Values of coefficients $\alpha$, $\beta$, $\gamma$, $\delta_1$, $\delta_2$ and $\delta_3$ for predicting each tensile property according to expressions 28 and 29. Coefficient $\delta_1$ for YS property is obtained as the average value from Equations (23) and (24).

| Property | $T_{aging}$ (°C) | $\alpha$ | $\beta$ | $\gamma$ | $\delta_1$ | $\delta_2$ | $\delta_3$ | $SDAS_0$ (μm) | $Mg_0$ (wt.%) | Equations |
|---|---|---|---|---|---|---|---|---|---|---|
| UTS (MPa) | | 33.624 | 121.141 | −1.534 | 225.057 | — | — | | | (10), (16), (22) |
| YS (MPa) | 165 | 59.746 | −103.322 | −1.255 | 212.934 | — | — | 28.4 | 0.36 | (11), (17), (23), (24) |
| A (%) | | −4.315 | 35.460 | −0.277 | — | −0.078 | 5.000 | | | (12), (18), (25) |
| UTS (MPa) | | 27.361 | 139.977 | −0.741 | 225.057 | — | — | | | (13), (19), (22) |
| YS (MPa) | 155 | 49.111 | −71.643 | −0.706 | 212.934 | — | — | 28.4 | 0.36 | (14), (20), (23), (24) |
| A (%) | | −3.844 | 33.230 | −0.286 | — | −0.176 | 29.485 | | | (15), (21), (26) |

Validation of Equation (9) (prediction of QI) and of the models included in Table 11 were made by using the tensile data from five and two different areas located on castings SK-7 and SK-8, respectively. As indicated in Section 2, these castings were produced with different Mg contents and then heat-treated at different aging temperatures and times. Thus, sets of 5 SK-7 and SK-8 castings were tested per Mg content and aging condition and the results of all tensile experiments performed are listed in Table 12.

The validation work is composed of two steps: prediction of QI values and then of tensile properties in each selected area of castings SK-7 (five areas) and SK-8 (two areas) following the models described in Equations (28) and (29), and in Table 11.

**Table 12.** Average values of UTS, YS and A for each Mg content and aging temperature and time in castings SK-7 and SK-8 (standard deviation values are shown between brackets).

| Mg (wt.%) | $T_{aging}$ (°C) | $t_{aging}$ (min) | Area | UTS (MPa) | YS (MPa) | A (%) |
|---|---|---|---|---|---|---|
| | | | SK-7 castings | | | |
| 0.34 | 165 | 280 | A1 | 312.5 (11.0) | 240.2 (2.0) | 11.6 (0.5) |
| | | | A2 | 311.4 (9.1) | 243.3 (3.7) | 11.5 (0.8) |
| | | | A3 | 316.5 (4.3) | 241.8 (4.0) | 11.1 (1.6) |
| | | | A4 | 310.1 (1.5) | 236.5 (2.9) | 10.0 (1.0) |
| | | | A5 | 309.1 (2.1) | 233.8 (2.3) | 8.3 (1.2) |
| | | 360 | A1 | 323.5 (2.2) | 252.1 (3.3) | 11.3 (0.5) |
| | | | A2 | 323.6 (3.0) | 257.0 (6.5) | 11.1 (0.4) |
| | | | A3 | 324.4 (2.7) | 252.3 (4.1) | 12.3 (0.7) |
| | | | A4 | 309.1 (2.1) | 251.3 (1.4) | 9.9 (1.3) |
| | | | A5 | 311.4 (2.5) | 250.6 (4.9) | 9.2 (0.7) |
| 0.42 | 165 | 280 | A1 | 330.9 (4.1) | 258.0 (6.9) | 10.3 (0.4) |
| | | | A2 | 332.9 (2.2) | 260.6 (5.0) | 10.1 (0.4) |
| | | | A3 | 333.1 (3.4) | 258.9 (4.0) | 10.5 (0.5) |
| | | | A4 | 325.9 (5.2) | 244.5 (3.2) | 9.0 (0.3) |
| | | | A5 | 324.7 (3.3) | 245.8 (5.5) | 8.1 (0.2) |
| | | 360 | A1 | 337.2 (1.7) | 266.6 (6.3) | 9.8 (0.7) |
| | | | A2 | 336.7 (1.8) | 266.0 (3.1) | 9.8 (1.4) |
| | | | A3 | 333.1 (1.5) | 257.8 (1.5) | 10.8 (1.3) |
| | | | A4 | 325.0 (3.1) | 261.0 (7.1) | 8.0 (0.7) |
| | | | A5 | 318.5 (3.0) | 256.1 (6.3) | 7.1 (0.3) |
| | | | SK-8 castings | | | |
| 0.30 | 155 | 360 | A1 | 292.4 (4.0) | 201.2 (3.2) | 13.5 (0.6) |
| | | | A2 | 288.1 (3.0) | 203.0 (2.3) | 14.0 (0.9) |
| 0.31 | | | A1 | 296.3 (2.1) | 202.3 (2.2) | 13.3 (1.1) |
| | | | A2 | 296.3 (3.3) | 203.2 (4.3) | 13.1 (1.4) |
| 0.33 | | | A1 | 297.3 (1.7) | 204.0 (2.4) | 12.5 (1.1) |
| | | | A2 | 296.7 (2.1) | 206.0 (1.6) | 12.9 (0.5) |
| 0.34 | | | A1 | 302.0 (6.0) | 212.0 (4.0) | 12.2 (0.2) |
| | | | A2 | 304.5 (5.5) | 209.5 (3.5) | 12.4 (0.8) |

For the first step QI values determined by using Equation (9) are compared to those obtained with Equation (8) in the four graphs of Figure 18. On the other hand, four different SDAS parameters were included in Equation (9) to determine the calculated QI values: the one experimentally measured on each area of castings SK-7 and SK-8 (see Table 8) and the ones determined by of Equations (5)–(7). QI data calculated with experimental values of SDAS and with those from Equation (6) are shown in the graphs included in Figure 18. QI values obtained with Equation (8) were considered as reference in these graphs.

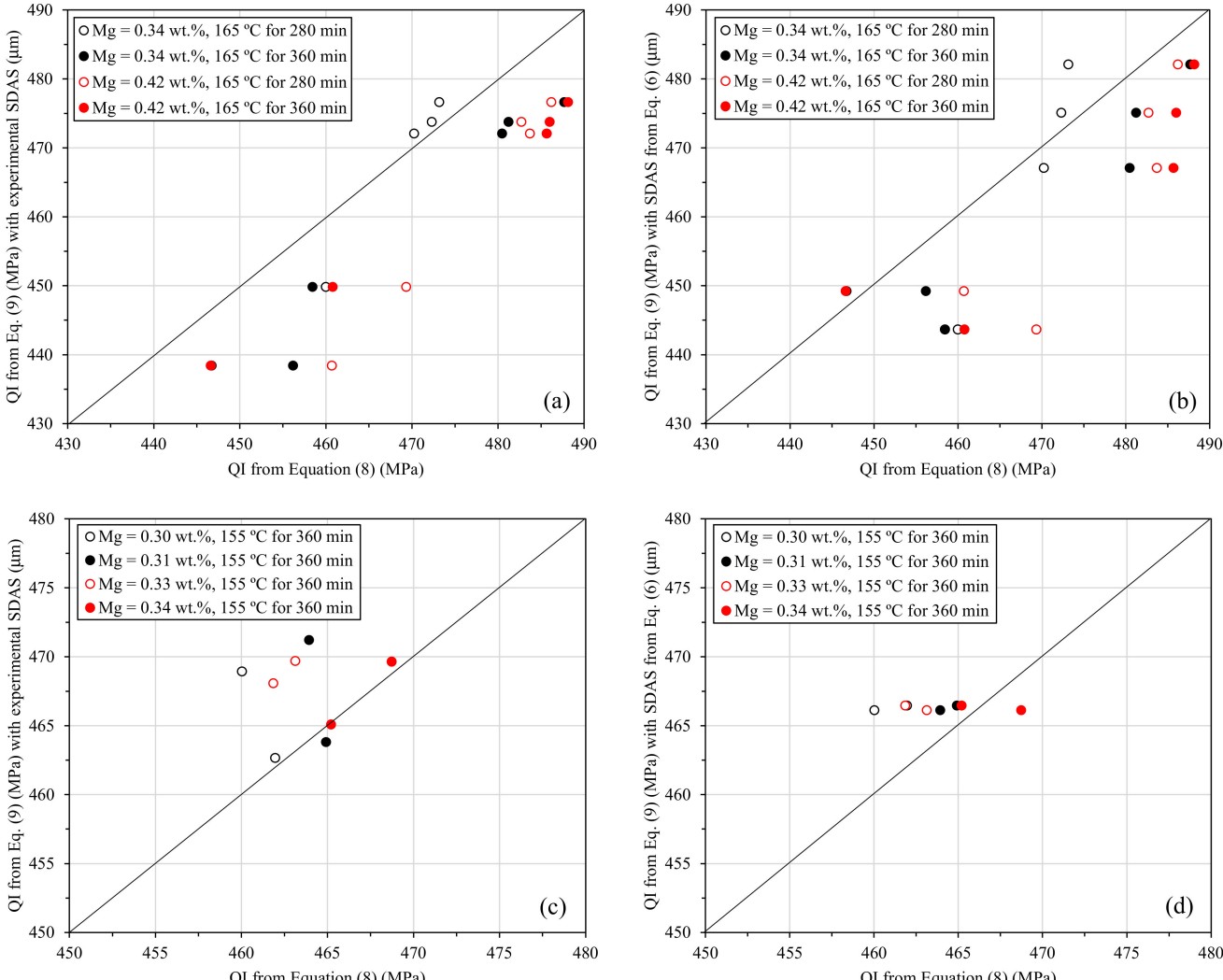

**Figure 18.** Comparison of QI values calculated with Equation (9) and those determined by using Equation (8). For the former equation experimentally measured SDAS (**a**,**c**) and the values of this parameter determined with Equation (6) (**b**,**d**) were used. The two graphs in the top row correspond to casting SK-7 while the two ones in the bottom row were obtained with data from castings SK-8. In case of SK-7 castings black and red circles correspond to the castings manufactured with 0.34 wt.% Mg and 0.42 wt.% Mg, respectively. In these two cases, open and solid symbols represent those castings aged at 165 °C for 280 min and at 165 °C for 360 min, respectively. For SK-8 castings increasing Mg content is given by black open (0.30 wt.%), black solid (0.31 wt.%), red open (0.33 wt.%) and red solid circles (0.34 wt.%). Solid lines are the bisectors.

The quantification of predictions is assessed using the absolute relative difference (RD) according to Equation (30). In this expression $M_{exp}$ means the experimental or reference value of the analyzed parameter and $M_{cal}$ is its calculated or predicted value.

$$RD\ (\%) = \left| \frac{(M_{exp} - M_{cal})}{M_{exp}} \cdot 100 \right| \tag{30}$$

In this case $M_{exp}$ are QI values determined using Equation (8) and $M_{cal}$ are QI values determined with Equation (9) using the experimentally measured SDAS and those calculated by Equations (5)–(7). Among data obtained from SK-7 castings, the maximum RD value is 5.7%, being the average values of this relative difference in the range of 1.1–3.1%. For the SK-8 castings, these two values are 2.5% and 0.1–2.1%. As expected, no significant

differences are found from data obtained at different aging treatments (SK-7 castings) thus confirming that QI is not affected by aging conditions.

It is also observed in Figure 18 that an increase of Mg content slightly increases those QI values calculated with Equation (8) (red dots appear moved to the right side in the four graphs) due to the hardening effect of this element. As expected, QI values determined through Equation (9) do not show this hardening effect as they were obtained using SDAS as the input parameter. Due to that, RD values show small increases with Mg content though this effect is not considered as relevant.

For the second step of validation, comparisons of experimental UTS, YS and A values (see Table 12) and those predicted using the models described in Equations (28) and (29), and in Table 11 are shown in the graphs of Figures 19–24. In these graphs, predictions made using experimental SDAS values and those calculated with Equation (6) are shown. The rest of the predictions obtained by using Equations (5) and (7) to determine SDAS lead to similar results as indicated in Table 13 which lists the maximum and average RD values obtained for each tensile property. In this case, $M_{exp}$ means experimental UTS, YS or A data while $M_{cal}$ represents the corresponding predicted values of these parameters.

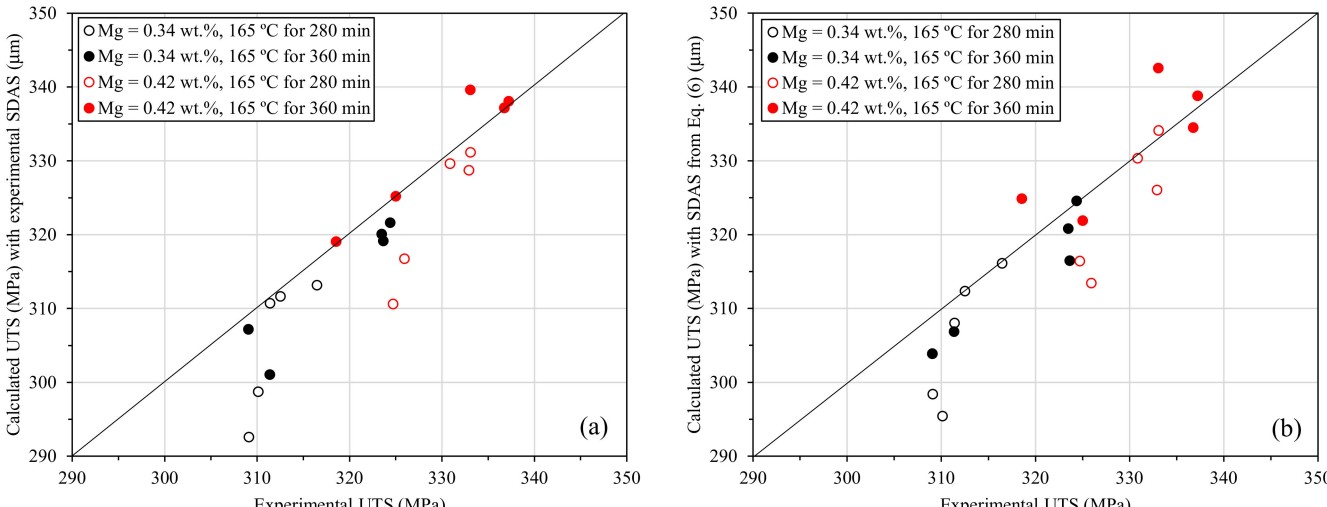

**Figure 19.** Comparison of experimental UTS values obtained from the SK-7 castings and those predicted by using Equation (28) and data included in Table 11. Two different SDAS parameters are used: experimentally measured SDAS (**a**) and the values of this parameter determined with Equation (6) (**b**). Black symbols correspond to SK-7 castings manufactured with 0.34 wt.% Mg while red symbols correspond to castings produced with 0.42 wt.% Mg. In both cases, open and solid circles represent those castings aged at 165 °C for 280 min and for 360 min, respectively. Solid lines are the bisectors.

For UTS and YS, the comparison of experimental and calculated values originates distributions of dots which follow the diagonal included in the graphs with some scattering. RD data listed in Table 13 are intended to quantify these scattering showing average and maximum values lower than 3.5% and 7.6%, respectively, for predictions made on SK-7 castings. In case of SK-8 castings, average and maximum RD values remain lower than 2.8% and 4.2%, respectively. Hence a maximum difference of about 14–17 MPa is found between the experimental and the calculated values of UTS and YS for SK-7 castings, while this range is reduced to 6–9 MPa for SK-8 castings. Notice that Equations (5)–(7) lead to satisfactory results and valid predictions of UTS and YS can be performed in these cases.

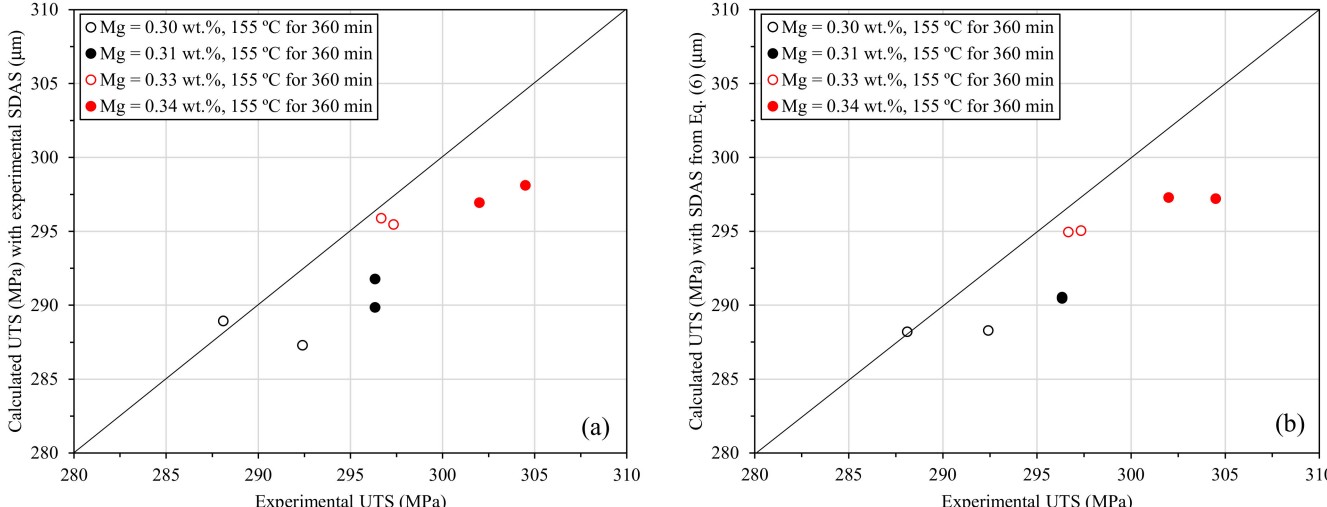

**Figure 20.** Comparison of experimental UTS values obtained from the SK-8 castings and those predicted by using Equation (28) and data included in Table 11. Two different SDAS parameters are used: experimentally measured SDAS (**a**) and the values of this parameter determined with Equation (6) (**b**). Black symbols correspond to SK-8 castings manufactured with 0.30 wt.% Mg (open circles) and with 0.31 wt.% Mg (solid circles) while red symbols correspond to castings produced with 0.33 wt.% Mg (open circles) and with 0.34 wt.% Mg (solid circles). All SK-8 castings were aged at 155 °C for 360 min. Solid lines are the bisectors.

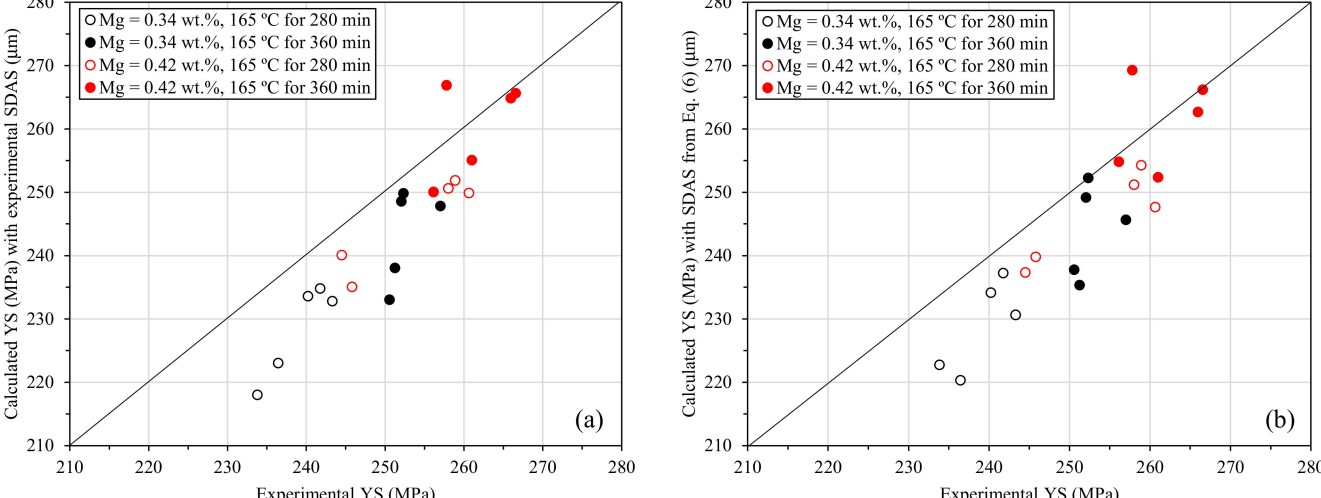

**Figure 21.** Comparison of experimental YS values obtained from the SK-7 castings and those predicted by using Equation (28) and data included in Table 11. Two different SDAS parameters are used: experimentally measured SDAS (**a**) and the values of this parameter determined with Equation (6) (**b**). Black symbols correspond to SK-7 castings manufactured with 0.34 wt.% Mg while red symbols correspond to castings produced with 0.42 wt.% Mg. In both cases, open and solid circles represent those castings aged at 165 °C for 280 min and for 360 min, respectively. Solid lines are the bisectors.

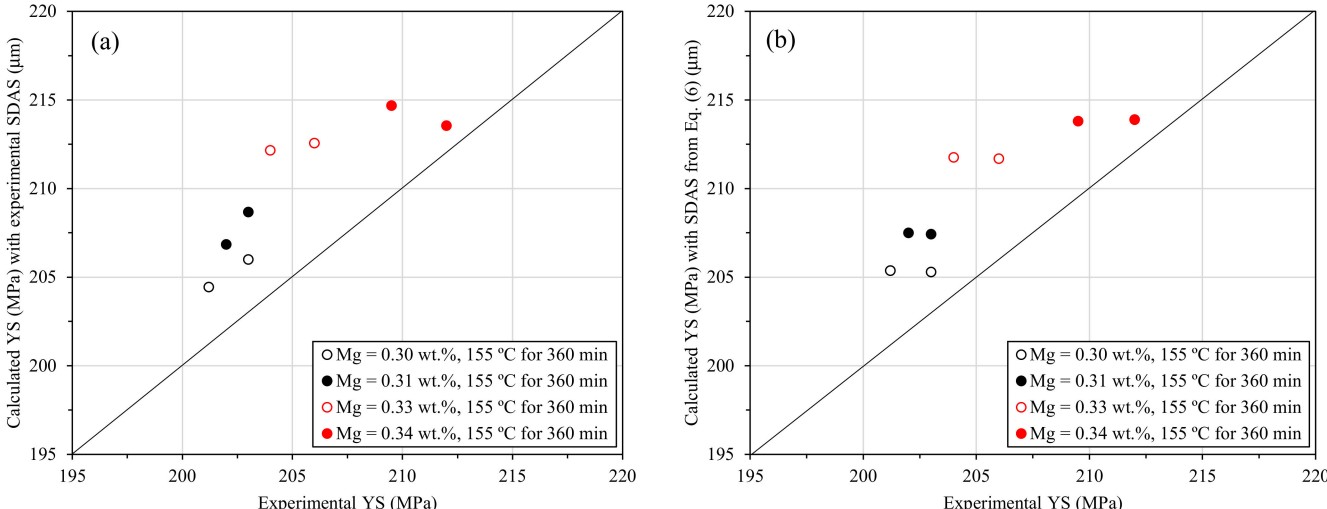

**Figure 22.** Comparison of experimental YS values obtained from the SK-8 castings and those predicted by using Equation (28) and data included in Table 11. Two different SDAS parameters are used: experimentally measured SDAS (**a**) and the values of this parameter determined with Equation (6) (**b**). Black symbols correspond to SK-8 castings manufactured with 0.30 wt.% Mg (open circles) and with 0.31 wt.% Mg (solid circles) while red symbols correspond to castings produced with 0.33 wt.% Mg (open circles) and with 0.34 wt.% Mg (solid circles). All SK-8 castings were aged at 155 °C for 360 min. Solid lines are the bisectors.

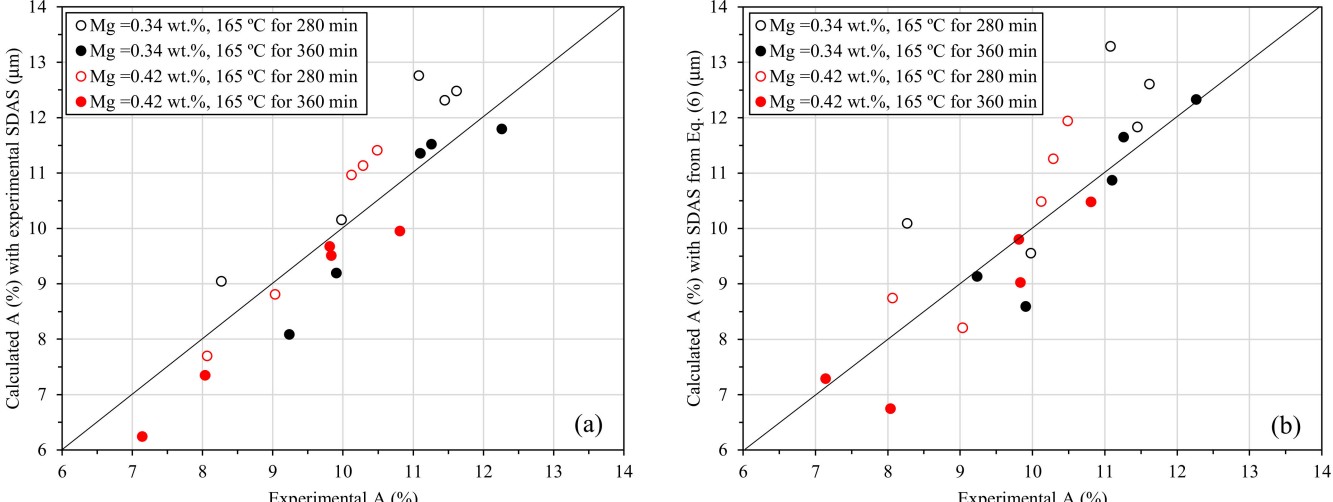

**Figure 23.** Comparison of experimental A values obtained from the SK-7 castings and those predicted by using Equation (29) and data included in Table 11. Two different SDAS parameters are used: experimentally measured SDAS (**a**) and the values of this parameter determined with Equation (6) (**b**). Black symbols correspond to SK-7 castings manufactured with 0.34 wt.% Mg while red symbols correspond to castings produced with 0.42 wt.% Mg. In both cases, open and solid circles represent those castings aged at 165 °C for 280 min and for 360 min, respectively. Solid lines are the bisectors.

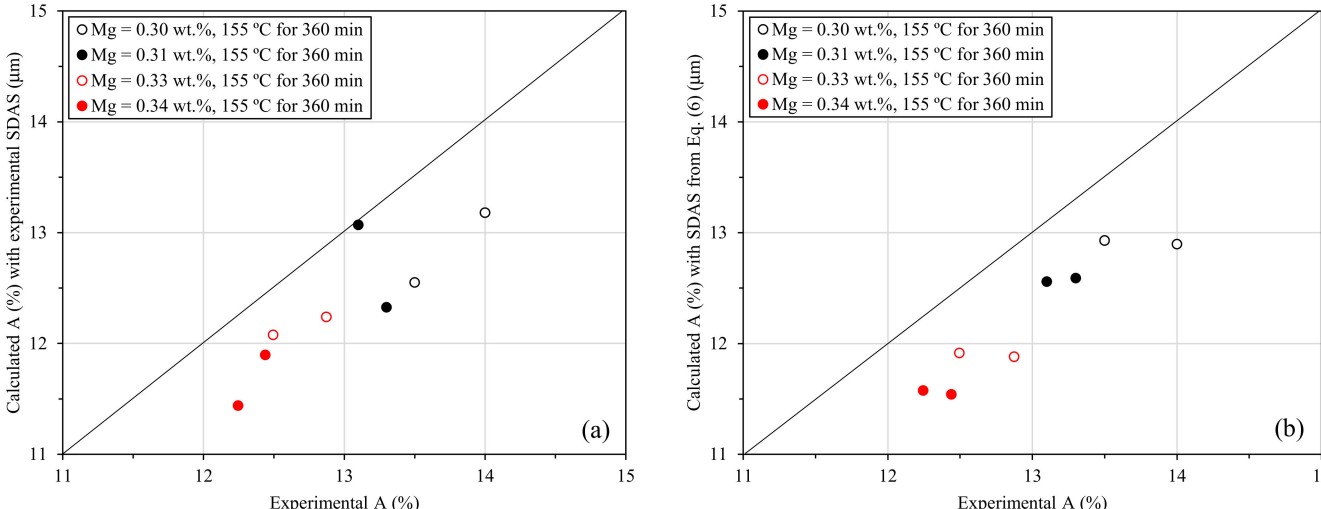

**Figure 24.** Comparison of experimental A values obtained from the SK-8 castings and those predicted by using Equation (29) and data included in Table 11. Two different SDAS parameters are used: experimentally measured SDAS (**a**) and the values of this parameter determined with Equation (6) (**b**). Black symbols correspond to SK-8 castings manufactured with 0.30 wt.% Mg (open circles) and with 0.31 wt.% Mg (solid circles) while red symbols correspond to castings produced with 0.33 wt.% Mg (open circles) and with 0.34 wt.% Mg (solid circles). All SK-8 castings were aged at 155 °C for 360 min. Solid lines are the bisectors.

**Table 13.** Maximum and average RD values (%) obtained when comparing experimental UTS, YS and A data with the corresponding predicted ones.

| SDAS | UTS | | YS | | A | |
|---|---|---|---|---|---|---|
| | **Maximum** | **Average** | **Maximum** | **Average** | **Maximum** | **Average** |
| SK-7 | | | | | | |
| Experimental | 5.3 | 1.5 | 7.5 | 3.2 | 15.1 | 6.8 |
| Equation (5) | 4.9 | 1.6 | 7.5 | 3.2 | 23.4 | 8.0 |
| Equation (6) | 4.7 | 1.6 | 6.8 | 3.1 | 22.1 | 7.6 |
| Equation (7) | 4.6 | 1.8 | 6.4 | 3.3 | 14.0 | 7.6 |
| SK-8 | | | | | | |
| Experimental | 2.2 | 1.3 | 3.9 | 2.3 | 7.3 | 5.0 |
| Equation (5) | 2.3 | 1.3 | 3.7 | 2.3 | 7.0 | 5.1 |
| Equation (6) | 2.4 | 1.3 | 3.8 | 2.2 | 7.9 | 5.8 |
| Equation (7) | 1.9 | 1.1 | 4.1 | 2.7 | 3.9 | 2.6 |

The hardening effects originated by increasing Mg content and by using longer aging time are also observed in Figures 19–22. On one hand, red solid symbols which represent those castings produced with the highest Mg contents stand at the highest UTS and YS values and there is a reduction of these properties when decreasing the content of this element. This fact is observed for both SK-7 and SK-8 castings though it is especially clear in case of the last group in which all castings were aged using the same conditions (155 °C for 360 min).

The effect of the longest aging time on UTS and YS can be observed in the graphs included in Figures 19 and 20 as the castings SK-7 were aged at 165 °C for 280 min (open circles) and 360 min (solid circles). Notice also that each group of symbols used in these two figures are composed by three dots located at high UTS or YS values (areas A1, A2 and

A3) and other two dots below them (areas A4 and A5). This behavior is due to the high SDAS values obtained in areas A4 and A5 of SK-7 castings (see Table 8).

In case of elongation at rupture, the scattering of data shown in Figures 23 and 24 is comparatively higher than that found for UTS and YS. In terms of RD, elongation shows the highest values (Table 13) though they mean maximum and average absolute differences between experimental and predicted A values lower than 2.5% and 1.0%, respectively.

As expected, distributions of red circles (high Mg content) and black circles (low Mg content) in the graphs shown in Figures 23 and 24 do follow opposite trends to those indicated above for UTS and YS due to the hardening effect provoked by both Mg content and aging parameters. Despite this fact, it is observed that each group of symbols in Figure 23 (castings SK-7) is also composed of three dots located at high A values and other two dots below them. As already explained for UTS and YS, this behavior is due to the low SDAS values measured in areas A1, A2 and A3 compared to those found in areas A4 and A5 in SK-7 castings (see Table 8). This finding shows again the strong influence of SDAS on elongation leading to a different effect on this property than that found for hardening.

## 4. Conclusions

Tensile properties data from different areas of castings are necessary information for designing new prototypes and for routinary quality controls on castings produced in foundry plants. Hence different groups of test castings and industrial steering knuckles all of them manufactured with A356.2 alloys and produced by LPDC technology have been studied in this work to correlate SDAS with those characteristics of actual and simulated cooling curves. Three equations based on solidification time and cooling rate measured on simulated cooling curves have been obtained to determine SDAS and these results have been satisfactorily compared to the experimental data. The results obtained in this investigation have also revealed the important influence of casting and cooling conditions on SDAS for a given alloy composition. In the present work, these parameters have been adapted for LPDC though a similar study can be made for other technologies. Quality index has been also related to SDAS and an equation has been found to estimate this index using SDAS data instead of tensile properties.

Separated studies have been made to investigate the effects of SDAS (used here as the structural parameter of the A356 alloys) on aging temperature and time, and of Mg content on tensile properties. In all three cases, relevant influences have been found which led to hardening effects when increasing the aging temperature and the Mg content. This fact has originated increases of UTS and YS parameters and reductions in elongation at rupture for these two last parameters. However, it is observed that decreasing SDAS has shown increases of all tensile properties, including elongation, due to refinement of $\alpha$-Al dendrites and homogenization of Fe-rich and Si-rich phases present in the solidified alloys.

Results obtained in these separated studies have been finally combined to develop new models for predicting tensile properties in castings manufactured with A356 alloys. Experimental data about SDAS and tensile properties of seven different areas present in two steering knuckles have been used to validate the obtained models with positive results. UTS and YS are strongly influenced by all three SDAS, Mg content and aging temperature and time. However, elongation is mainly affected by SDAS parameter, with a likely small influence of the two aging times used in this work and a significant effect from Mg content.

The developed predictive models represented by Equations (28) and (29) together with data included in Table 11 are considered highly important tools as they can be used in production plants to estimate mechanical properties of existing castings or prototypes only considering the simulation results. Such early possibility minimizes high costly steps during prototypes development like the production of LPDC dies and their modifications during optimization work. This effective estimation of tensile properties also allows optimizing aspects related to T6 heat treatment parameters, die design and process parameters like number and location of cavities, design of cooling systems, mold thickness, temperature distribution on the cavities by activation and closing of cooling lines, pressure curves, etc.

These predictive models have been developed at composition ranges of 7.10–7.45 wt.% Si and 0.30–0.42 wt.% Mg (Fe and Mn contents lower than 0.10 wt.%), and cooling rates of 0.23–1.92°C/s for Sr-modified and $TiB_2$ grain refined A356 alloys. It is expected that the application range of the obtained models will be within the variable intervals indicated above though other intervals can be investigated.

**Author Contributions:** Conceptualization, J.A.S. and P.R.; methodology, J.A.S. and I.L.; software, A.R.; validation, J.A.S., J.S. and E.O.d.Z.; formal analysis, J.S. and J.A.S.; data curation, J.S.; writing—original draft preparation, J.S. and J.A.S.; writing—review and editing, J.S., J.A.S., E.O.d.Z. and P.R.; supervision P.R.; project administration, J.A.S. and P.R. All authors have read and agreed to the published version of the manuscript.

**Funding:** This research did not receive any external funding.

**Data Availability Statement:** Not applicable.

**Acknowledgments:** The authors thank the collaborating efforts made by Fagor Ederlan Group in sharing their knowledge regarding LPDC casting parameters and the support in performing T6 heat treatments. As well, it is noteworthy the efforts done by the staff of the LPDC Innovation Casting Cell and Materials Science Department located at Edertek.

**Conflicts of Interest:** The authors declare no conflict of interest.

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
