# Peer review of "Towards the Prediction of Tensile Properties in Automotive Cast Parts Manufactured by LPDC with the A356.2 Alloy"

_metals, doi:10.3390/met12040656_

Round 1
Reviewer 1 Report
The manuscript “Towards the prediction of tensile properties in automotive cast parts manufactured by LPDC with the A356.2 alloy” by J.A. Santamaría et al. studied a set of A356 alloys which had been prepared and then used to manufacture test castings and automotive castings in laboratory and industrial conditions. Test castings were used to predict secondary dendritic arm spacing (SDAS) by using thermal parameters obtained from experimental cooling curves.
It represents a good contribution for your valuable journal as it is Metals.
However, before the Editor makes a decision, I suggest that the authors must take into account the following corrections:
1. The authors would explain clearly in the abstract what is the novelty and what is the added value in this manuscript.
2. The authors would check typing errors throughout the manuscript.
3. English style should also be improved.
4. References are suggestive. I am convinced that it is useful for the manuscript if will be included in the References section following papers with the same topics or using similar procedures, ex.:
- Microstructure Evolution and Solidification Behavior of a Novel Semi- Solid Alloy Slurry Prepared by Vibrating Contraction Inclined Plate, Metals, 11, 1810, (2021), https://doi.org/10.3390/met11111810;
- Mechanics of Elastic Composites, Chapman & Hall/ CRC Press, U.S.A, 708 pp., (2003).
For these reasons, I recommend the acceptance of this manuscript for publication after Minor Revision.
Author Response
Please, see the attachment

Reviewer 2 Report
The experimental methods and results were very well organized.
This paper contributed to the development of a model for predicting SDAS values in industrial castings produced by the LPDC process. Also, it is evaluated as an excellent paper to calculate the physical properties of A356.2 alloy as variables of SDAS, Mg content, and T6 aging.
Author Response
Please, see the attachment

Reviewer 3 Report
In general, the study was carried out at a fairly high level. The amount of experimental work carried out is very large.
Of the small flaws, I can recommend indicating the scale bars in Figures 1 and 6. So that the reader can evaluate the dimensions of the mold and casting.
In Figure 14b, red arrows indicate Mg2Si inclusions. It would be nice to sign these inclusions in the figure in the same red color.
Author Response
Please, see the attachment
